# Functional multispectral optoacoustic tomography imaging of hepatic steatosis development in mice

Shan Huang[1,2], Andreas Blutke[3], Annette Feuchtinger[3], Uwe Klemm[1], Robby Zachariah Tom[4,5], Susanna M Hofmann[4], Andre C Stiel[1] ORCID & Vasilis Ntziachristos[1,2,*] ORCID

## Abstract

The increasing worldwide prevalence of obesity, fatty liver diseases and the emerging understanding of the important roles lipids play in various other diseases is generating significant interest in lipid research. Lipid visualization in particular can play a critical role in understanding functional relations in lipid metabolism. We investigated the potential of multispectral optoacoustic tomography (MSOT) as a novel modality to non-invasively visualize lipids in laboratory mice around the 930nm spectral range. Using an obesity-induced non-alcoholic fatty liver disease (NAFLD) mouse model, we examined whether MSOT could detect and differentiate different grades of hepatic steatosis and monitor the accumulation of lipids in the liver quantitatively over time, without the use of contrast agents, *i.e.* in label-free mode. Moreover, we demonstrate the efficacy of using the real-time clearance kinetics of indocyanine green (ICG) in the liver, monitored by MSOT, as a biomarker to evaluate the organ's function and assess the severity of NAFLD. This study establishes MSOT as an efficient imaging tool for lipid visualization in preclinical studies, particularly for the assessment of NAFLD.

**Keywords** biomarker; ICG; lipid metabolism; NAFLD; optoacoustic imaging
**Subject Categories** Digestive System; Metabolism; Methods & Resources

## Introduction

Non-alcoholic fatty liver disease (NAFLD) is characterized by excessive accumulation of lipids in the liver (Burt *et al*, 2015). Due to the increasing prevalence of obesity worldwide, NAFLD is rapidly becoming one of the leading causes of severe liver disease (Younossi *et al*, 2018). Currently, the assessment of NAFLD severity relies on histopathology, which cannot monitor pathological and functional changes during therapeutic intervention without liver biopsies, limiting its use in preclinical studies. Thus, there is a need for a sensitive, quantitative and non-invasive tool for NAFLD assessment in both preclinical and clinical settings, in order to enable longitudinal studies of potential therapies for this disease.

Ultrasonography (USG) is a widely used screening method for steatosis and fibrosis (Wong *et al*, 2018). However, it is not sensitive enough to detect mild steatosis. Recently, the use of magnetic resonance imaging to estimate the proton density fat fraction (MRI-PDFF) has been shown to more accurately and sensitively detect grades of steatosis in NAFLD than USG (Park *et al*, 2017). Moreover, MRI-PDFF is responsive to changes in liver fat over time (Noureddin *et al*, 2013). Despite its value in clinical trials and experimental studies, MRI-PDFF is limited in clinical settings by its high cost (European Association for the Study of the *et al*, 2016).

Currently, liver function assessment relies on blood tests for classic biomarkers that are poorly correlated with NAFLD development, such as aspartate aminotransferase (AST) and alanine aminotransferase (ALT) (Maximos *et al*, 2015). Beyond those, the indocyanine green (ICG) clearance test is considered a valuable method for dynamic assessment of liver function (Halle *et al*, 2014). ICG is a water-soluble fluorescent dye that is exclusively cleared from the blood via the liver. The clearance of ICG through the hepatobiliary tract depends on liver blood flow and hepatic and biliary function (Sakka, 2007). Thus, its clearance from the blood stream serves as a biomarker of liver function (Hoekstra *et al*, 2013), making the ICG clearance test potentially valuable for NAFLD monitoring (Seifalian *et al*, 2001) (Danin *et al*, 2018). However, the readout of clinical tests of ICG clearance from the blood stream cannot reflect the full process of ICG clearance in the liver. Using a rabbit model, Seifalian *et al* (2001) demonstrated that the ICG uptake rate correlated significantly with blood flow and microcirculation, while the excretion rate correlated with liver enzymes, which reflected the function of hepatocytes. Since NAFLD starts with lipid accumulation in hepatocytes, the early changes in liver function caused by NAFLD should affect the excretion of ICG from the hepatocytes to the bile ducts. Therefore, blood ICG clearance, which reflects mainly the uptake of ICG by the liver rather than the excretion of ICG from the hepatocytes, might not be sensitive enough for the detection of early changes in liver function during NAFLD development. This

1 Chair of Biological Imaging, School of Medicine, Central Institute for Translational Cancer Research (TranslaTUM), Technical University of Munich, Germany
2 Institute of Biological and Medical Imaging, Helmholtz Zentrum München (GmbH), Neuherberg, Germany
3 Research Unit Analytical Pathology, Helmholtz Zentrum München (GmbH), Neuherberg, Germany
4 Institute of Diabetes and Regeneration Research, Helmholtz Zentrum München (GmbH), Neuherberg, Germany
5 Medizinische Klinik und Poliklinik IV, Ludwig-Maximilians-Universität München, Munich, Germany
*Corresponding author. Tel: +49 89 4140 7211; Fax: +49 89 4140 6748; E-mail: v.ntziachristos@tum.de

insensitivity may also underlie the scepticism in some studies, as to whether the ICG clearance rate correlates with the severity of steatosis (Danin *et al*, 2018) (Lykke Eriksen *et al*, 2019) (Parker *et al*, 2017).

Multispectral optoacoustic tomography (MSOT) separates lipids from other tissue components based on their absorption spectra in the near-infrared (NIR) range. Lipid spectra have a characteristic peak at a narrow spectral range around approximately 930 nm (Jacques, 2013). The technique detects ultrasound waves that are generated by thermo-elastic expansion of tissue upon absorption of light by different light absorbing biomolecules. The technique combines the molecular sensitivity of light with the resolution of ultrasound, yielding a potent technology for the study of lipids. Due to the low attenuation of light in the NIR compared to the visible or infrared spectral ranges, operation in the NIR range allows light penetration through several centimetres of tissue, making MSOT highly effective for *in vivo* imaging (Reber *et al*, 2018) (Ntziachristos *et al*, 2019). Moreover, MSOT is sensitive to haemoglobins in the 700–900 nm range and to ICG at the dye's peak absorption at ˜800 nm, potentially allowing monitoring of lipids, ICG distribution and morphological features and dynamics imparted by haemoglobin contrast (Beziere *et al*, 2015) (Chen *et al*, 2018). Optoacoustic (OA) measurements (also known as photoacoustic) have been previously considered for investigating steatosis *in vivo*, but at the short wavelength infrared (SWIR) range, in particular at 1,220 and 1,370 nm (Rom *et al*, 2019) (Xu *et al*, 2016). These studies only allowed the generation of bulk signals with a pixelated (speckle) appearance, that did not offer an anatomical optoacoustic reference, since the wavelengths employed do not reveal haemoglobin contrast. Moreover, generally, attenuation of light in the > 1,200 nm region is higher to significantly higher than in the NIR. In contrast, our investigation herein focused on examining whether the 930 nm range would be appropriate for imaging liver lipids and steatosis. Imaging in the NIR comes with several advantages as it can be seamlessly integrated with morphological and functional images obtained at the 700–900 nm range afforded by the same tunable laser and can yield higher sensitivity and specificity for the detection and quantification of lipids by avoiding the strong water absorption in the > 1,200 nm range.

We hypothesized that the spectral range of 930 nm could be used to non-invasively evaluate steatosis by directly imaging and quantifying changes in hepatic lipid content of mice *in vivo*. As a first step, we used MSOT to characterize the lipid content, as well as water, oxy- and deoxy-haemoglobin (HbO$_2$ and Hb), within various tissues *in vivo*, including liver, kidney, brown adipose tissue (BAT), white adipose tissue (WAT), aorta and Sulzer vein (SV). We then assessed the ability of MSOT to detect excessive lipid in the liver of a NAFLD mouse model *in vivo* with quantitative readouts and compared our data to semi-quantitative histopathological grading, the current gold standard for evaluation of hepatic steatosis. The performance of our MSOT-based method in predicting grades of hepatic steatosis was demonstrated by analysing the receiver operating characteristic (ROC) curve. We further monitored the progression of steatosis in mice over time using our MSOT imaging approach. Lastly, we tested the feasibility of assessing hepatic function by detecting ICG clearance kinetics based on OA detection of ICG directly in the liver. We show elaborate optoacoustic images of the underlying tissues and the ability of the spectral range around 930nm to be employed as a novel imaging tool for NAFLD assessment in preclinical studies, with future potential applications in the clinical monitoring of NAFLD, especially due to the lower attenuation of light in the NIR versus SWIR.

# Results

## MSOT imaging of liver, kidney, adipose tissues and blood vessels in non-obese mice

First, we examined the ability of MSOT to differentiate various key tissues based on the spectral signatures of HbO$_2$, Hb, lipid and water in healthy mice, which is the prerequisite for tissue identification and eventual pathophysiological observation. Hb and HbO$_2$ can be used to assess metabolic metrics, such as total blood volume (TBV) and oxygenation of the tissue (sO$_2$). The right kidney serves as a negative control for the quantification of fat in the liver, as it is located close to the liver and does not accumulate large amounts of fat during obesity-related development of hepatic steatosis. Interscapular brown adipose tissue (iBAT) and retroperitoneal white adipose tissue (rpWAT) were positive controls for the lipid detection. The aorta and SV were also examined, mainly for tissue oxygenation analysis due to their pure blood content and natural difference in sO$_2$. We determined the region of interest (ROI) within the liver, kidney, iBAT, rpWAT, aorta and SV in mice by inspecting the anatomical and spectral features of certain tissues in single wavelength OA images at 800 nm and corresponding images with unmixing data of tissue contents (Fig 1A). The OA features of the tissues were then evaluated by inspecting their absorption spectra (Fig EV1A–C) in comparison with individual endogenous absorbers (Fig EV1D). To enhance the visualization of spectral features, we normalized all tissue spectra to the highest OA signal acquired in the 700–960 nm range (Fig 1B). Since the *in vivo* spectra of deep tissue can be affected by light attenuation from the surrounding tissues, we compared *in vivo* spectra of liver, kidney, iBAT and rpWAT with their *ex vivo* spectra from the same animal. As shown in Fig 1C and Appendix Fig S1, the *ex vivo* spectra of liver, iBAT and kidney have a sharper descending slopes in general, which reflects the reduction in sO$_2$ in these blood-rich tissues. The rpWAT spectra remained largely the same due to a low blood content compared to the other three tissues. In general, the spectral signatures of tissues, e.g. the characteristic lipid peak at 930 nm in iBAT and rpWAT, were mostly unaltered.

In the following quantitative analysis, we normalized the tissue TBV and sO$_2$ to the respective aorta readouts of each mouse. As expected, liver and kidney were detected as blood-rich tissues (Fig 1D and E). Due to the double vascularization system in the liver, which contains 80% venous blood from the portal vein (Herzog *et al*, 2012), the sO$_2$ in liver is significantly lower than that in kidney (*P* < 0.0001). In comparison with rpWAT, iBAT has higher blood content (*P* < 0.0001; Fig 1E), which reflects its high vascular density (Sibulesky, 2013).

To test the feasibility of using MSOT to quantitatively isolate excessive lipid from liver tissue, we analysed a phantom with lipid fractions ranging from 0 to 100 % (Fig EV2). At 930 nm, the MSOT signal intensity increased with increasing lipid fraction in the phantom (Spearman *r* = 0.98, *P* < 0.0001; Fig EV2A and B). Linear unmixing readouts of lipid showed excellent correlation with the

lipid fraction (Fig EV2C; Spearman $r = 1$, $P < 0.0001$). We also analysed phantoms with homogenized liver tissue and lipid mixed in different ratios. According to histopathological quantification and MRI-PDFF data from other studies, lipid fractions in steatotic livers rarely exceed 60% (Tang *et al*, 2013; Bannas *et al*, 2015). We therefore tested a series of liver tissue phantoms with lipid fractions from 0 to 60% (Fig EV3). The lipid signature absorption peak at 930 nm became more prominent with increasing lipid fraction in the phantom (Fig EV3A and B). The MSOT lipid readouts showed excellent correlation with the lipid fraction (Spearman $r = 1$, $P < 0.0001$; Fig EV3C).

These initial results indicated that MSOT can analyse the blood and fat contents of tissues such as liver to reveal their biological properties.

## MSOT imaging of steatosis in mice with diet-induced obesity

After demonstrating MSOT's ability to analyse tissue content in healthy animals, we assessed whether pathological changes related to fat accumulation alter the spectral signature of the tissues. We chose the diet-induced obesity mouse model which develops steatosis along with obesity (Nnodim & Lever, 1988). The mice fed with a high-fat diet (HFD) had enlarged fat depots and significantly higher body weight ($P = 0.005$; Fig 2A and B). While no significant change was observed in the concentration of Hb and $HbO_2$ in any of the tissues analysed, the lipid content significantly increased in the liver ($P = 0.004$) and iBAT ($P = 0.007$; Fig 2C). This increase indicates a synergistic malfunction in lipid metabolism in these tissues, consistent with similar findings from recent studies (Niu *et al*, 2019) (Sharma *et al*, 2019). The metabolic status of the tissues revealed by $sO_2$ and TBV was not altered dramatically in the obese mice (Fig 2D). In the *ex vivo* spectra of liver and kidney, more steeply descending slopes are expected between 700 to 800 nm compared to the *in vivo* spectra, due to a decrease in $sO_2$ after sacrificing the animal. These results suggest that MSOT can assess pathological changes associated with steatosis, without labels.

## Specific and sensitive detection and quantification of lipid in liver by MSOT

Here, we verify the specificity and sensitivity of lipid detection in normal and steatotic livers by MSOT. In addition to the linear unmixing method, we introduce a simpler analysis method for label-free lipid detection, referred to herein as "difference analysis". The readout from this method is calculated by subtracting the optoacoustic intensity at 930 nm from that at 700 nm and dividing that value by the intensity at 800 nm (difference analysis readout = $I_{700} - I_{930}/I_{800}$). Since the intensity a 930 nm varies with lipid content, while the value at 700 nm stays relatively constant because it is related to blood content, the difference value $I_{700} - I_{930}$ should be inversely proportional to lipid concentration. The intensity at 800 nm was used to normalize the difference value because oxygenation saturation does not affect the readout from this wavelength as it is the oxy-deoxy isosbestic point. For performance analysis, we divided difference analysis readout from kidney by that from liver to further eliminate the individual variance while achieving positive relevance with steatosis. The final readout is named as difference index. The main advantage of using such a simple

readout for lipid content is that it allows a fast data acquisition and analysis as well as an economic light source with only three wavelengths.

To demonstrate the efficacies of linear unmixing and difference analysis for quantifying liver lipid content, we first imaged the livers of one lean mouse and one obese mouse with extreme steatosis (confirmed by Oil Red O lipid staining) both *in vivo* and *ex vivo* (Fig 3A and B). The tissue spectra of the steatotic liver showed strong signals at 930 nm (indicative of lipid) when compared to the control, both *in vivo* and *ex vivo* (Fig 3C). As expected, the spectra of the kidneys *in vivo* in the same subjects were nearly identical (Fig 3C). The difference value $I_{700} - I_{930}$ was lower in the steatotic liver than in the normal liver, both *in vivo* (Fig 3B, bottom) and *ex vivo* (Fig 3D), in accordance with the high lipid content of the steatotic liver. Furthermore, we quantified the lipid content from 6 healthy and 9 NAFLD mice with severe steatosis. The average lipid contents calculated by linear unmixing were 1.29 a.u. and 674.6 a.u. ($P = 0.0004$) in the control and steatotic livers, respectively, while the corresponding difference value was 0.772 and 0.396 ($P = 0.0008$; Fig 3E). Both linear unmixing readout and difference value from the adjacent right kidney were unaffected by the hepatic steatosis (Fig 3E).

Furthermore, we examined the performance of linear unmixing and difference analysis for lipid quantification and compared the results with histopathological grading. Representative examples of the full range of grades of steatosis are shown in Fig 4A. Apart from grading, we also performed an automated quantification of lipid using HE staining (Fig EV4). The mean areas of lipid for grade 0, 1, 2 and 3 livers were 0.04, 0.90, 2.17 and 15.30, respectively (grades 0 versus 1: $P < 0.0001$; grades 1 versus 2: $P = 0.0007$; and grades 2 versus 3: $P = 0.0086$). By linear unmixing of the spectra measured by MSOT, the difference between grades 3 and grade 0 was highly significant ($P < 0.0001$) (Fig 4B). Moreover, linear unmixing is also able to distinguish grade 1 or 2 and grade 3 steatosis (grades 1 versus 3: $P < 0.0001$ and grades 2 versus 3: $P = 0.0012$) (Fig 4B). The histological quantification and linear unmixing readouts of lipid showed strong correlation (Spearman $r = 0.82$, $P < 0.0001$; Fig EV5A). Livers with all grades of steatosis had significantly higher difference index values than the normal liver (grades 0 versus 1: $P = 0.0007$; grades 0 versus 2: $P = 0.0009$; and grades 0 versus 3: $P < 0.0001$) (Fig 4C). However, difference index values were not distinguishable from each other for grades 1, 2 and 3. There was no strong correlation between linear unmixing readouts and difference analysis readouts (Spearman $r = 0.55$, $P = 0.0001$; Fig EV5B).

To investigate the prediction accuracy of MSOT readouts for steatosis in preclinical setting, we calculated the area under the receiver operating characteristic (AUROC) and the potential cutoff value for assessing different grade of steatosis (Fig 4D; Table 1). The ROC analysis results suggested that both linear unmixing and difference analysis can predict grades of steatosis with excellent (AUROC > 0.8) to outstanding (AUROC > 0.9) discriminatory ability (Mandrekar, 2010). Direct comparison showed that linear unmixing was more accurate than difference analysis at predicting grade 2–3 steatosis ($P = 0.008$, Fig 4D).

We further tested the possibility of tracking the progression of steatosis *in vivo*. HFD-fed mice gradually accumulated excessive lipids in fat depots (Fig 4E). Despite individual variations in actual values, the measured hepatic lipid levels all exhibited an upward trend during the 3 months of HFD feeding, indicating a gradual

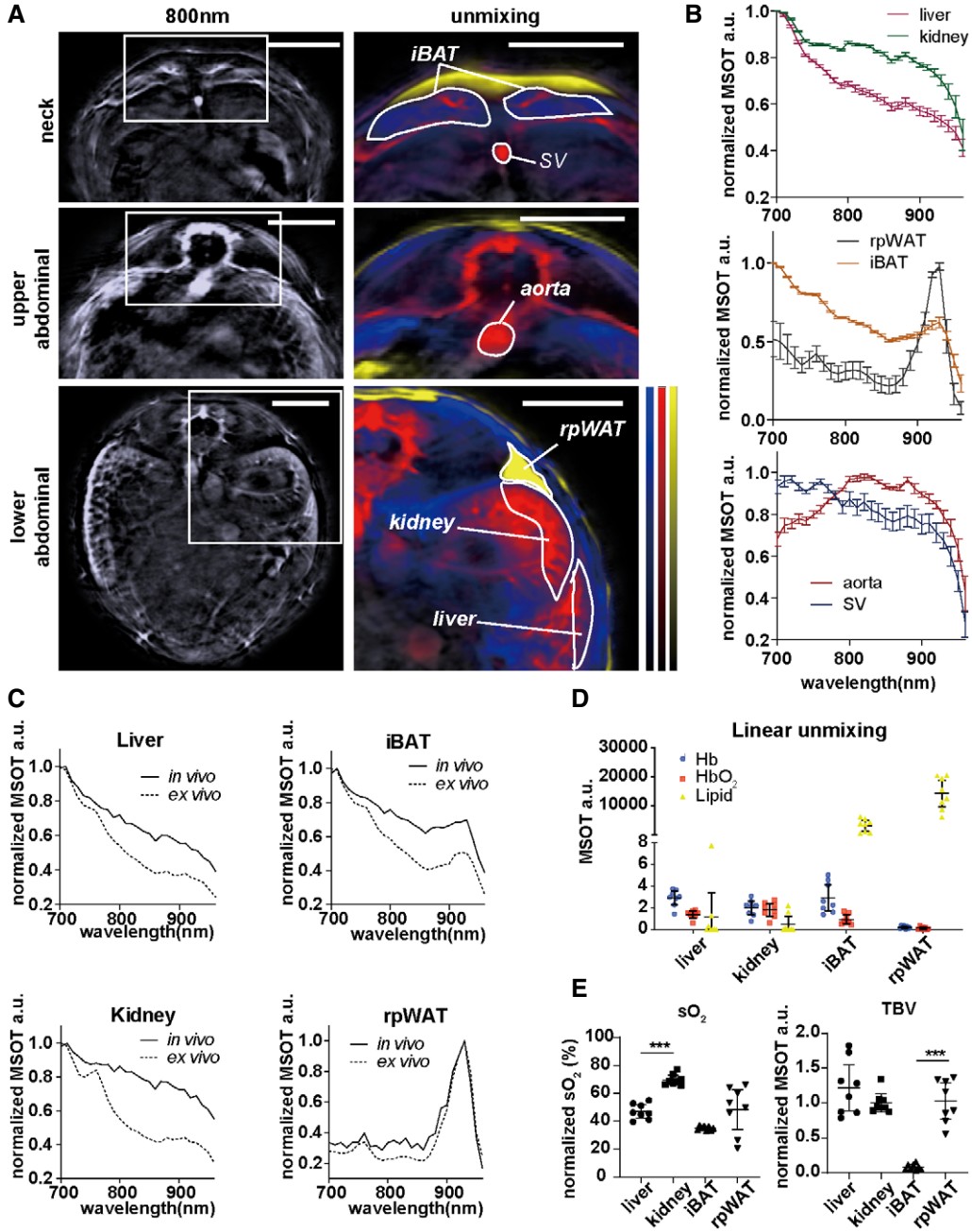

**Figure 1. MSOT imaging of liver, kidney, interscapular brown adipose tissue (iBAT), retroperitoneal white adipose tissue (rpWAT), aorta and Sulzer vein (SV) from healthy mice *in vivo* and *ex vivo*.**

A   Reconstructed MSOT images (800 nm) without and with linear unmixing data in the neck and the upper and lower abdominal areas, showing the ROIs for the liver, kidney, iBAT, rpWAT, aorta and SV. Unmixing result: blue for Hb, red for HbO$_2$ and yellow for lipid. The colour bar shows the colour coding of MSOT a.u. from 0 to maximum (bottom to top) (maximum value neck/upper abdominal/lower abdominal: Hb: 2.5/3.5/0.9; HbO$_2$:2.9/4.9/2.5; lipid: 24000/19000/13000). Scale bar: 4 mm.

B   Normalized spectra of liver, kidney, iBAT, rpWAT, aorta and SV. Data represent the mean ($\pm$ 95% confidence) from 8 animals ($n = 8$).

C   Normalized spectra of liver, kidney, iBAT and rpWAT *in vivo* and *ex vivo*. The two spectra in each plot are from the same animal.

D, E   Linear unmixing results from liver, kidney, iBAT and rpWAT. Each dot represents data from one animal, in total 8 animals ($n = 8$). Data represent the mean ($\pm$ 95% confidence). The unpaired *t*-test was used to verify the statistical significance. SO$_2$ liver versus kidney: $P = 1.66E-07$; TBV iBAT versus rpWAT: $P = 5.58E-07$.

Data information: In the figure, A.U. = arbitrary units.

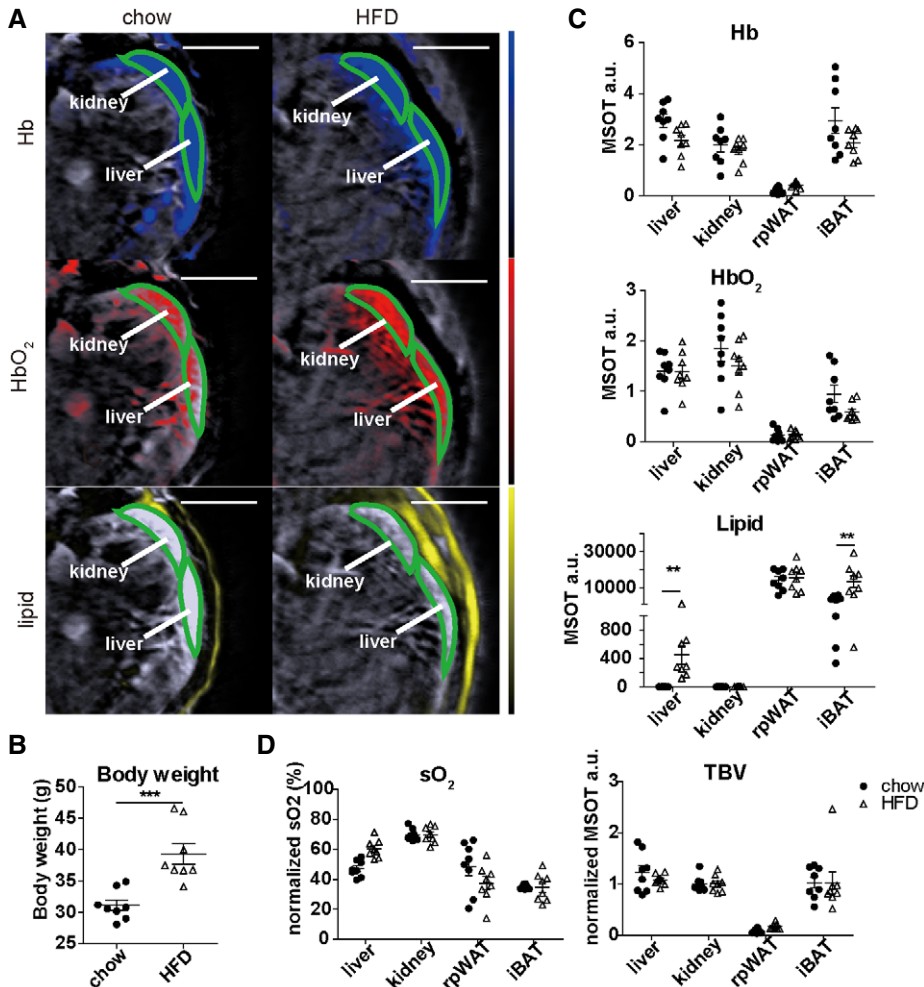

**Figure 2. Comparison of oxygenation and lipid content of target tissues between healthy and obese mice using MSOT imaging.**

A   Reconstructed MSOT image (800 nm) with linear unmixing data of Hb, HbO₂ and lipid from chow- and HFD-fed mice. The colour bar shows the colour coding of MSOT a.u. from 0 to maximum (bottom to top) (maximum value chow/HFD: Hb: 1.3/0.6; HbO2: 1.8/0.9; lipid: 27000/27000). Scale bar: 4 mm.

B   Body weight of chow- and HFD-fed mice. Data represent the mean (± 95% confidence) from 8 animals ($n = 8$). The unpaired $t$-test was used to verify the statistical significance. $P = 0.0005$.

C   Unmixing result of Hb, HbO₂ and lipid from liver, kidney, iBAT and rpWAT. Each dot represents data from one animal, in total 8 animals ($n = 8$). Data represent the mean (± 95% confidence). The unpaired $t$-test was used to verify the statistical significance. Lipid chow versus HFD liver: $P = 0.3339$; iBAT: $P = 0.0068$.

D   Tissue oxygenation (sO₂) and total blood volume (TBV) results from liver, kidney, iBAT and rpWAT. Each dot represents data from one animal, in total 8 animals ($n = 8$). Data represent the mean (± 95% confidence). The unpaired $t$-test was used to verify the statistical significance.

Data information: In the figure, A.U. = arbitrary units.

accumulation of lipid in liver over time (Fig 4F). ND fed mice did not accumulate lipid over time. These results suggested the feasibility of monitoring steatosis progression by MSOT in a preclinical setting.

**Functional OA imaging of liver with ICG**

Here, we tested our hypothesis that hepatic ICG clearance could serve as a biomarker for liver function in the assessment of NAFLD in mice. After administration of ICG, we traced its levels based on linear unmixing in healthy and steatotic livers for 2 h, before which the normal liver clearance of ICG was completed (Fig 5A, Appendix Fig S2A, Movie EV1). The liver spectra showed peak absorption at 800nm in both control and steatotic liver, indicating existence of ICG in the tissue after injection. The 800nm peak disappeared in the control liver 2 h after ICG injection, while in steatotic liver, the peak remained visible throughout the observation (Fig 5B, Appendix Fig S2B). We further monitored the kinetics of ICG signal for 1 h after ICG administration in 5 normal and 5 NAFLD mice with grade 3 steatosis, confirmed by histology and MSOT imaging ($P < 0.0001$) (Fig 5C). The steatotic livers showed slower elimination of ICG than the normal liver (Fig 5D), indicating an impaired excretion function. Consistently, there was more residual ICG signal in the steatotic liver than in the normal liver under fluorescent microscopy *ex vivo* ($P = 0.0079$) (Fig 5E, Appendix Fig S3).

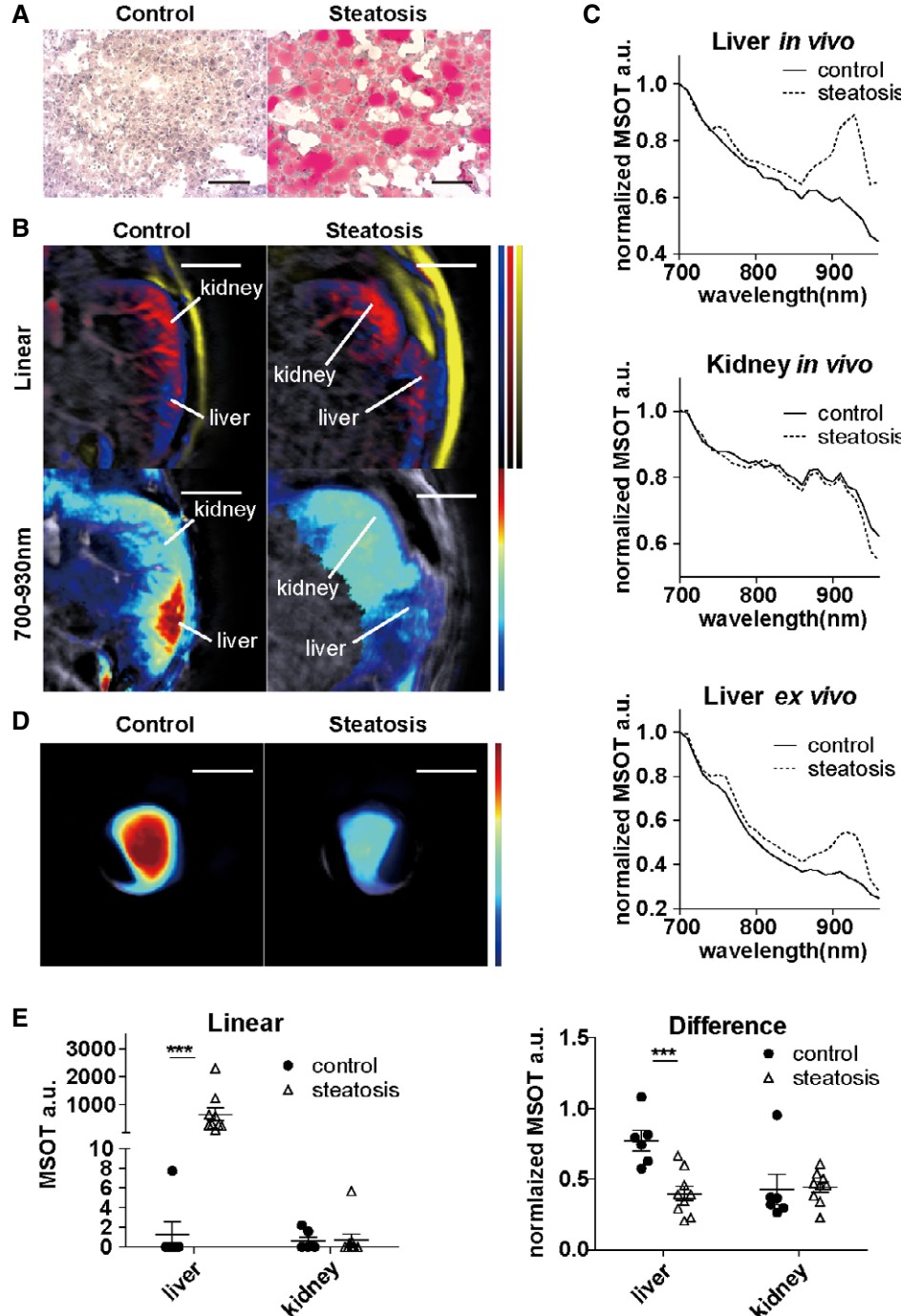

**Figure 3. MSOT detection of lipid content in healthy and steatotic livers *in vivo* and *ex vivo*.**

A  Oil Red O staining of control (healthy) and steatotic liver. Scale bar: 100 μm.

B  Reconstructed MSOT image (800 nm) with linear unmixing and difference value of lipid in lower abdominal section. Unmixing result: blue for Hb, red for HbO2, yellow for lipid and jet for 700–930 nm difference. The colour bar shows the colour coding of MSOT a.u. from 0 to maximum (bottom to top) (maximum value control/steatosis: Hb: 1.1/0.5; HbO2: 1.9/0.8; lipid: 21,000/24,000; 700–900 nm: 3,000/3,000). Scale bar: 4 mm.

C  Normalized spectra of livers and kidneys from a control (healthy) and a subject with hepatic steatosis. Each spectrum is from the same animal.

D  Reconstructed MSOT image (800 nm) with difference values of lipid in control (healthy) and steatotic liver ex vivo. The colour bar shows the colour coding of MSOT a.u. from 0 to maximum (bottom to top) (maximum value control/steatosis: 10,000/10,000). Scale bar: 4 mm.Data in panel A–D are from the same subjects.

E  Linear unmixing readouts of lipids and difference values from control and steatotic (grade 3) livers. Control: n = 6, steatosis: n = 9. Each dot represents data from one animal (control: n = 6, steatosis: n = 9). Data represent the mean (± 95% confidence). The Mann–Whitney test and the unpaired t-test were used to verify the statistical significance in linear and difference data, respectively. Linear liver control versus steatosis: P = 0.0004; difference liver control versus steatosis: P = 6.62E-06.

Data information: In the figure, A.U. = arbitrary units.

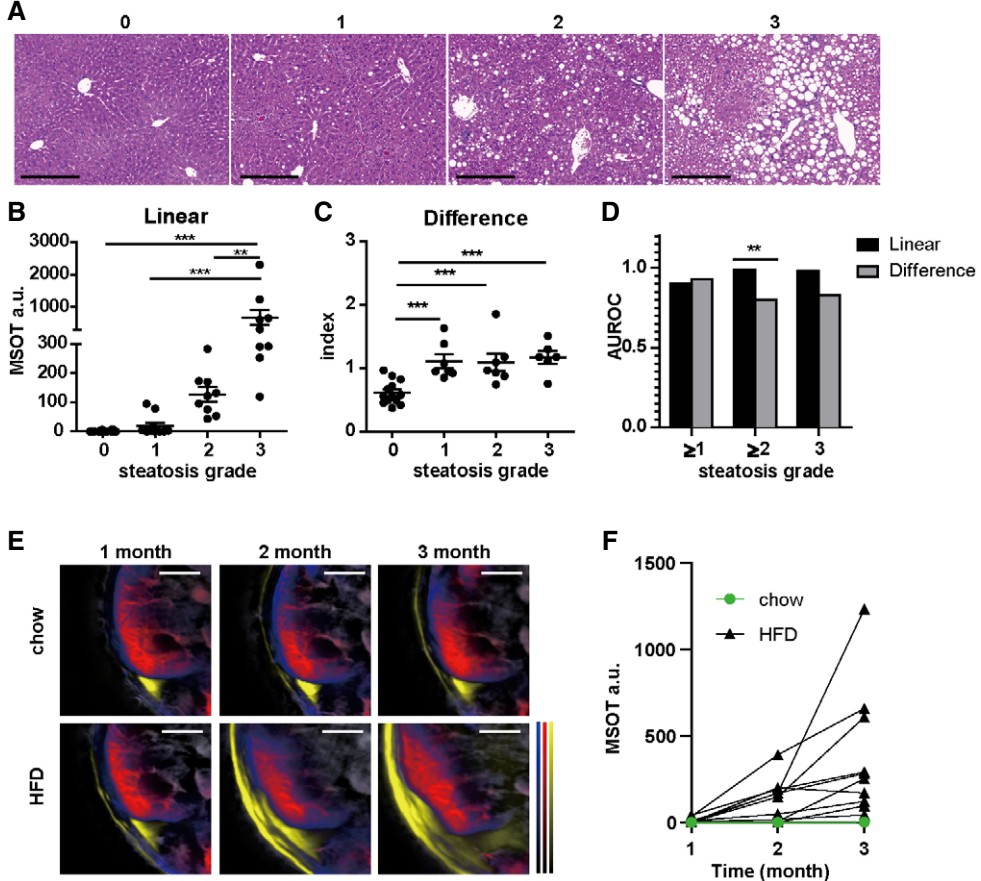

**Figure 4. Quantitative assessment of steatosis by MSOT in comparison with histological grading.**

A  HE staining of livers with grade 0 − 3 steatosis. Scale bar: 100 μm.

B  Quantification of linear unmixing of lipid in liver. Each dot represents data from one animal (grade 0: $n = 14$; grade 1: $n = 11$; and grades 2 and 3: $n = 9$). Data represent the mean (± 95% confidence). The Mann–Whitney test was used to verify the statistical significance. Grade 0 versus grade 3: $P = 1.22E-06$; grade 1 versus grade 3: $1.19E-05$; and grade 2 versus grade 3 $P = 0.0012$.

C  Quantification of difference analysis readouts in liver. Same subjects as in B. Each dot represents data from one animal (grade 0: $n = 14$; grade 1: $n = 11$; and grades 2 and 3: $n = 9$). Data represent the mean (± 95% confidence). The unpaired *t*-test was used to verify the statistical significance. Grade 0 versus grade 1: $P = 9.93E-05$; grade 0 versus grade 2: $P = 0.0004$; and grade 0 versus grade 3: $P = 4.90E-05$.

D  Diagnostic accuracy of linear unmixing and difference analysis in diagnosing dichotomized grades of steatosis. AUROC was calculated from data points in panel B and C. The *z* test was used for comparisons of AUROC between 2 groups. Grade 2 linear versus difference: $P = 0.008$.

E  Reconstructed MSOT image (800 nm) with linear unmixing data of Hb, HbO₂ and lipid from chow- and HFD-fed mice. Unmixing result: blue for Hb, red for HbO₂ and yellow for lipid. The colour bar shows the colour coding of MSOT a.u. from 0 to maximum (bottom to top) (maximum value chow 1 month/ chow 2 months/ chow 3 months/HFD 1 month/HFD 2 months/HFD 3 months: Hb: 2.5/1.9/2.9/2.7/1.8/1.3; HbO2: 2.8/2.6/2.8/3.1/2.1/1.9; lipid: 27,000/19,000/29,000/32,000/26,000/27,000). Scale bar: 4 mm.

F  Longitudinal track of lipid in mouse livers. Each line represents data from one animal. Each data point is from a single measurement of each animal at each time point. All animals have grade 3 steatosis at the endpoint confirmed by histology.

Data information: In the figure, A.U. = arbitrary units.

# Discussion

Few studies use label-free OA imaging methods to investigate the biology and pathology of the liver (Rom *et al*, 2019) (Xu *et al*, 2016), most likely due to the high absorption of light by blood and the tissue depth. Herein, we demonstrate that by exploiting its multispectral nature, MSOT imaging is able to detect and separate all grades of steatosis with quantitative readouts, supporting its value for disease progression and treatment efficacy monitoring in a preclinical setting. Compared to linear unmixing, difference analysis

showed similar sensitivity as linear unmixing towards mild steatosis, despite its use of only three wavelengths (700, 800 and 930 nm). For a conceivable clinical translation, this approach would be faster, more straight-forward to analyse, while requiring less cost-intensive laser sources than linear unmixing, making it suitable for screening and early detection of steatosis.

In this work, we have also introduced hepatic ICG clearance as a biomarker for NAFLD assessment in mice using time-lapse MSOT imaging. With the time-resolved data acquired by MSOT, we were able to differentiate the kinetics of ICG uptake and

**Table 1. Prediction accuracy of linear unmixing and difference analysis in detecting each grade of steatosis.**

| Cross-sectional steatosis grade classification (n = 43) | | Grades 1–3 (n = 29) versus grade 0 (n = 14) | Grades 2–3 (n = 18) versus grade 0–1 (n = 25) | Grade 3 (n = 9) versus grades 0–2 (n = 34) |
|---|---|---|---|---|
| Linear unmixing | cutoff | 17.3 | 42.9 | 252.9 |
| | AUROC (95%CI) | 0.90 (0.81–0.98) | 0.99 (0.96–1.01) | 0.98 (0.94–1.02) |
| | Se | 72% | 100% | 89% |
| | Sp | 100% | 92% | 97% |
| | PPV | 100% | 90% | 89% |
| | NPV | 64% | 100% | 97% |
| Difference analysis | cutoff | 0.74 | 0.97 | 1.01 |
| | AUROC (95% CI) | 0.93 (0.86–1.02) | 0.80 (0.67–0.94) | 0.83 (0.68–0.98) |
| | Se | 100% | 67% | 89% |
| | Sp | 79% | 84% | 82% |
| | PPV | 91% | 75% | 57% |
| | NPV | 100% | 78% | 97% |
| Difference analysis versus linear unmixing P value (sample size not enough) | | 0.48 | 0.008 | 0.06 |

Se, sensitivity; Sp, specificity; NPV, negative predictive value; and PPV, positive predictive value.

excretion in control and steatotic livers in mice. A similar method using tissue NIR spectroscopy has been introduced as a comprehensive liver function test, but its application was restrained by the invasive procedures including an open surgery (Shinohara et al, 1996). Our non-invasive method can overcome this limitation while providing equivalent data. Since ICG is a proven contrast agent for clinical use, it is feasible to test our MSOT-based ICG detection method in a clinical setting. In future work, we will investigate the connection between ICG kinetics in liver and pathological changes, severity and prognosis of the disease for clinical translation.

Our study reveals some limitations of MSOT-based steatosis assessment. First, although the numeric readout of MSOT correlates to different grades of steatosis with proper thresholds, it cannot be converted into an absolute lipid fraction, despite its continuity. The main reasons for this are as follows: (i) Heterogeneity of tissues and body composition *in vivo*. The unmixing readout is calculated from the absorption spectrum, which is greatly affected by the tissue that surrounds the ROI. Tissue contents and body composition vary from subject to subject, especially in the context of metabolic disease. This leads to an individualized problem that no unmixing algorithm can yet solve. In the future, developing an AI-based method could address this problem. (ii) Lack of co-registration of MSOT ROI and sample ROI for the validation assay. Since MSOT can only analyse a limited volume in the liver due to the depth, a co-registration of MSOT and the validation assay that provides the ground truth lipid fraction readout is indispensable for the conversion of MSOT a.u. into absolute lipid fraction. It is challenging whether the validation assay is an *ex vivo* method. One possibility is to co-register MSOT and MRI-PDFF images. However, an advanced algorithm that solves the tissue heterogeneity problem discussed in 1) is still a prerequisite for this conversion. A second major limitation revealed in our study is the imaging depth, which is not sufficient for analysing a whole liver section. This is mainly due to the spectrum corruption in deep

tissue. Since liver is rich in blood, a strong absorber for the wavelengths we used, the excitation light is quickly exhausted in liver tissue. Therefore, we need to avoid selecting deep voxels with corrupted spectra by using a two-step method for ROI selection described in the Materials and Methods section. However, this method is time-consuming and not completely objective. To improve the ROI selection for future studies, it would be useful to develop an automated method using deep learning, which could cluster spectra of specific tissues and correct the corrupted spectra until the signals drop beyond an auto-determined threshold. Another technical issue is water as coupling media could diminish the sensitivity of lipid detection by MSOT, as water absorbs at wavelengths longer than 900 nm. Using other coupling media, such as heavy water, can eliminate the influence of water absorption and increase the sensitivity of lipid detection by MSOT.

Despite this study aiming only at the analysis of steatosis in mice, the encouraging results in the use of MSOT in measuring spectral data in humans (Reber et al, 2018) suggest the possibility of a clinical translation of the method described in this work. In contrast to MRI, clinical MSOT is a portable modality, which could enable bed-side or outpatient monitoring of NAFLD development. However, this method needs to be further adapted for clinical translation. First, since we were using preclinical small-animal MSOT throughout this study, the method introduced here needs to be adjusted for the clinical version of MSOT, which is a handheld 2D or 3D imaging modality (Reber et al, 2018). Therefore, suitable imaging position(s) need to be defined and standardized, along with imaging procedures to ensure efficient visualization and analysis of the liver in humans and low intra- and inter-observer variability. This is particularly important when using MSOT for monitoring disease development or therapeutic response. Second, the desired imaging depth in humans, which is commonly several centimetres, could present a challenge to preserve the sensitivity and the capability for quantification by our lipid detection method, especially in morbidly obese patients who have thick subcutaneous

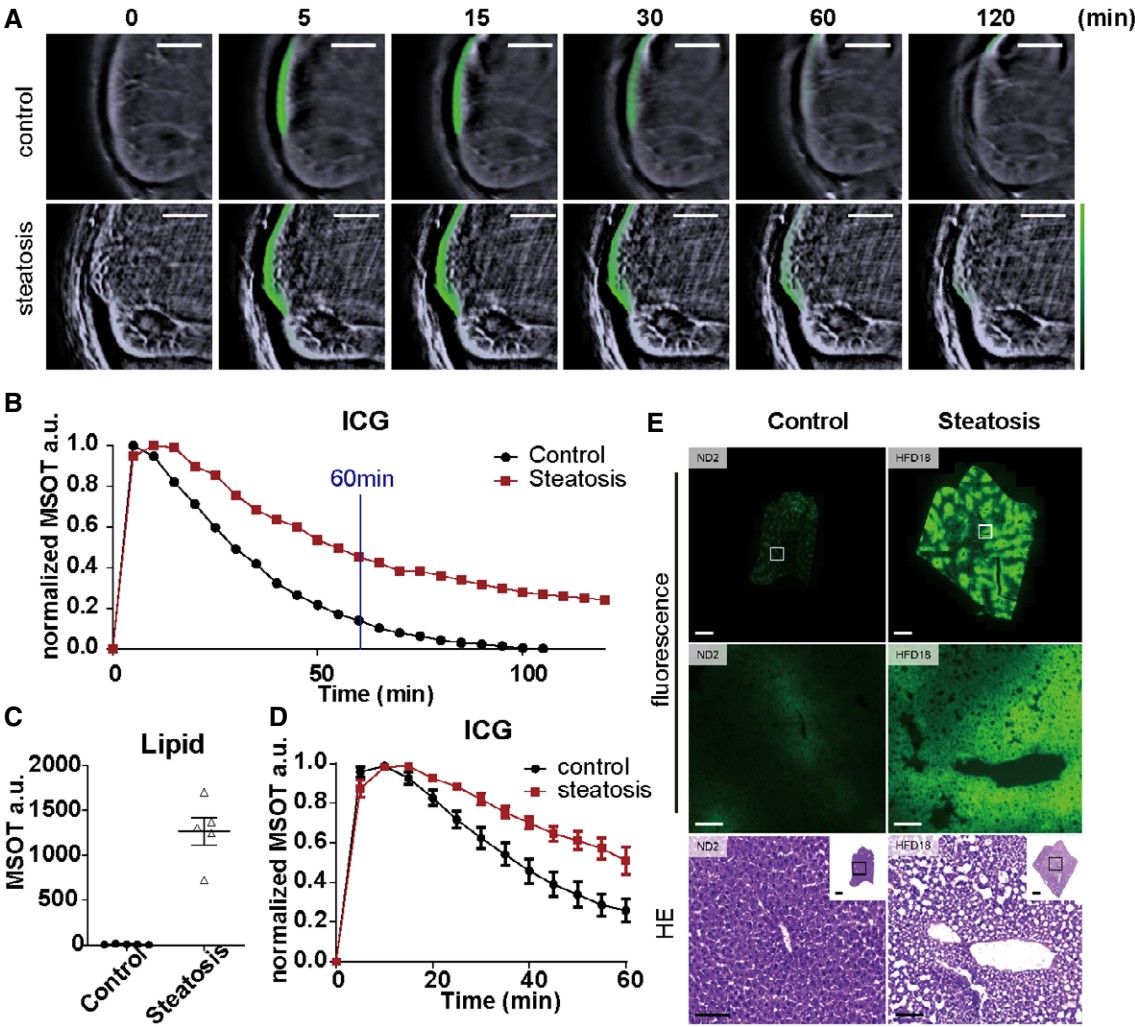

**Figure 5. MSOT imaging of ICG clearance in mouse livers.**

A Reconstructed MSOT image (800 nm) with linear unmixing data of ICG from mice with and without steatosis. The colour bar shows the colour coding of MSOT a.u. from 0 to maximum (bottom to top) (maximum value control/steatosis: 0.07/0.02). Scale bar: 4 mm.

B Normalized spectra of control and steatotic liver 10 and 120 min after ICG injection. Each line represents data from one animal. Each data point is from a single measurement of each animal at each time point.

C Quantification of linear unmixing of lipid in livers with and without steatosis. Each dot represents data from one animal (*n* = 5). Data represent the mean (± 95% confidence).

D Longitudinal monitoring of ICG intensity in control and steatotic livers (time interval: 5 min; total length: 60 min). Each data point represents the mean from 5 animals at each time point. For each curve, data are normalized to the highest intensity acquired.

E Representative fluorescence microscopy and HE staining images of control and steatotic liver. Scale bar: 100 µm.

Data information: In the figure, A.U. = arbitrary units.

fat, which diminishes the light energy reaching the liver. Intravascular imaging of lipids, which has been demonstrated in many studies for plaque detection (Wu *et al*, 2017) (Choi & Mandelis, 2019), might allow access to the periportal area of the liver through catheter-based OA imaging. This would overcome the limitation of imaging depth and potentially lead to a more sensitive method for steatosis detection. However, intravascular imaging using MSOT includes an invasive procedure, which would limit its routine clinical use.

Clinical applications of optoacoustic imaging are rapidly emerging, with studies suggesting the potential of MSOT for diagnosing skin tumours (Chuah *et al*, 2017), breast cancer (Shiina *et al*, 2018), vascular malformations (Masthoff *et al*, 2018a), systemic sclerosis (Masthoff *et al*, 2018b), Crohn's disease (Knieling *et al*, 2017) and muscular dystrophy (Regensburger *et al*, 2019). Here, we introduce MSOT as a new tool for NAFLD assessment in preclinical settings. Outcomes of future clinical trials may confirm the potential of MSOT as a sensitive, quantitative and cost-effective imaging method for NAFLD assessment, which will ease the monitoring of the disease, as well as promote its therapeutic study for better healthcare outcomes.

 

# Materials and Methods

### Animals

Animals were kept at 24 ± 1°C and on a 12:12-h light–dark cycle with free access to food and water. 2-4 mice were kept in each cage. To induce obesity and NAFLD, 8- to 10-week-old male C57BL/6BrdCrHsd-Tyrc mice (Janvier Labs, France; The Jackson Laboratory, USA) were fed with HFD comprising 58% kcal from fat (D12331; Research Diet, New Brunswick, NJ, USA) for up to 6 months. Control mice from the same strain were fed with standard rodent diet (Altromin 1314, Altromin Spezialfutter GmbH & Co, Germany). 14 mice per sex were used in each group (56 mice in total). 48 out of 56 animals yielded experimental data. The mice used in the ICG experiment were injected with 2.5 µg/g body weight of ICG (ICG-Paulsion von Verdye 25 mg, Diagnostic Green, Germany and USA) in 100 µl saline intravenously through a catheter. 6 mice per group were used for the ICG study. 10 out of 12 animals yielded experimental data. The average weights of the healthy and NAFLD mice were 36.6 and 46.5 g, respectively. The same mice were used for imaging and histopathological analysis. After the final imaging time point, the imaged animal was sacrificed and the liver was isolated and divided into formalin-fixed, paraffin-embedded (FFPE) and frozen parts for *ex vivo* analysis. The animal studies were approved and conducted in accordance to the Animal Ethics Committee of the government of Upper Bavaria, Germany.

### Multispectral optoacoustic tomography

MSOT measurement were conducted with a 256-channel real-time imaging MSOT scanner (inVision 256, iThera Medical, Germany) described before (Razansky *et al*, 2011; Reber *et al*, 2018) (Dima *et al*, 2014). For both *in vivo* and *ex vivo* measurements, imaging was performed at 27 wavelengths in the range of 700–960 nm in steps of 10 nm. At each wavelength, 10 frames were averaged per section during data acquisition. By employing measurements in the NIR range, we only required a single fast tunable laser that can switch between wavelengths at 50 Hz rates.

For *in vivo* measurements, mice were anaesthetized by continuous inhalation of 2% isoflurane (vaporized in 100% oxygen at 0.8 l/min) and subsequently placed within an animal holder in a supine position relative to the transducer array. Ultrasound gel (Ultrasound gel AQUASONIC clear, Parker Laboratories, Fairfield, USA) was used as couplant between the polyethylene membrane and the animal. Mice were then placed onto a thin, clear, polyethylene membrane and positioned in the water bath in a sample holder as described earlier (Razansky *et al*, 2011). The total duration of anaesthesia was about 1 h for the NAFLD experiment and about 2.5 h for the ICG experiment. All MSOT imaging experiments were carried out at 34°C. The respiration rate of each mouse was monitored during imaging and kept in the range of 40–60 breaths per min.

For *ex vivo* measurement, tissues were isolated from the mice immediately after euthanasia. Samples were placed into 1-ml syringes filled with PBS and held in position in the MSOT by a custom designed holder.

### Phantom measurements

For the lipid phantom measurements, duck fat was mixed with chloroform in different ratios (see Fig EV2) and loaded into 1-ml syringes, with the tips sealed. The syringes (the phantoms) were held in position in the MSOT by a custom designed holder. For the liver/lipid tissue phantom measurements, non-steatotic veal liver (confirmed by histology) was homogenized and mixed with duck fat in different ratios. The tissue mixture was sealed in a cylinder-shaped hole in the middle of an agar phantom. The phantom was held in position in the MSOT by a custom holder.

### Histopathology

Liver tissue specimens were sampled according to established organ sampling and trimming guidelines for rodent animal models (Ruehl-Fehlert *et al*, 2003). The samples were fixed in neutrally buffered 4% formaldehyde solution for 24 h and subsequently embedded in paraffin. Sections (3 µm thick) were stained with haematoxylin and eosin (HE), using a HistoCore SPECTRA ST automated slide stainer (Leica, Germany) with prefabricated staining reagents (HistoCore Spectra H&E Stain System S1, Leica, Germany), according to the manufacturer's instructions. Histopathological examination was performed by a pathologist in a blinded fashion (*i.e.* without knowledge of the treatment-group affiliations of the examined slides). Histopathology was carried out on random liver regions. One section (average area: 35.31 mm$^2$) per liver was analysed for each animal. Hepatic steatosis was graded semi-quantitatively (Finan *et al*, 2016) according to the proportion of liver tissue in the sections affected by the presence of fat vacuoles in hepatocyte profiles (grade 0: < 5%; grade 1: 5–33%; grade 2: 33–66%; and grade 3: > 66%).

Additionally, the amount of lipid vacuoles in the liver sections were morphometrically determined by automatic digital image analysis of scanned slides (Sachs *et al*, 2020) (Axio Scan.Z1 scanner equipped with 20x objective, Zeiss, Germany) using the commercially available software Definiens Developer XD 2 (Definiens AG, Germany). Entire liver sections with an average area of 35.31 mm$^2$ were analysed. A specific rule set was defined in order to detect and quantify the lipid vacuoles of the hepatocytes based on morphology, size, pattern, shape, surroundings and colour. The calculated parameter was the percentage of surface area considered as lipid vacuoles divided by the total surface area of the whole analysed tissue section for each slide.

### Oil Red O staining

Livers were isolated immediately after euthanasia. Tissues were embedded in O.C.T. (Tissue-Tek; Sakura Finetek, USA) and frozen to −50°C before cryoslicing in steps of 10 µm at −20°C using a modified cryotome (CM 1950; 839 Leica Microsystems, Germany). The sections were air-dried for 2 h before staining. The sections were first fixed in neutrally buffered 4% formaldehyde solution, briefly washed with running tap water for 1–10 min and rinsed with 60% isopropanol. The sections were then stained with Oil Red O working solution and freshly prepared according to the manufacturer's manual (O0625-25G, Sigma-Aldrich, Merck KGaA, Darmstadt, Germany) for 15 min followed by rinse with 60% isopropanol. The nuclei were then lightly counterstained with alum

haematoxylin. The sections were finally rinsed with distilled water before mounting of coverslips with aqueous mountant (P36971, Thermo Fisher Scientific, Waltham, Massachusetts, USA).

## Data analysis

MSOT data were analysed by ViewMSOT software (v3.8, iThera Medical, Munich, Germany). MSOT images were reconstructed using the model linear method. "Linear regression" method was used for unmixing of Hb, HbO$_2$, Lipid, H$_2$O and ICG. For lipid detection, an additional method was also applied, termed "difference analysis", which is based on a simple calculation of readout from 700, 800 and 930 nm (see Result section "*Specific and sensitive detection and quantification of lipid in liver by MSOT*"). The data analysis also included calculation of the total blood volume (TBV = HbO$_2$ + Hb, displayed as arbitrary units) and the oxygen saturation of blood in the tissue (sO$_2$ = HbO2 / TBV, displayed as percentage). For ROI selection, a preliminary ROI based on anatomical information was drawn on the image reconstructed from data at 800 nm (the oxy-deoxy isosbestic point). The final ROI was fine-tuned on the Hb unmixing image by evaluating whether the observed absorber distribution was reasonable based on prior knowledge of the target tissue; for example, a blood-rich tissue such as liver or kidney should have a relatively homogenous distribution of Hb. Each unmixing data point was averaged from three ROIs in the same subject for statistical analysis. To normalize the ICG intensity data, we subtracted the background signal of the ROI in the liver from the raw unmixing data before ICG administration and then normalized the data to the peak value.

## *Ex vivo* quantification of ICG tracer

Dissected liver samples were snap-frozen in liquid nitrogen, cryosectioned at 12 μm (CM1950, Leica Microsystems, Wetzlar, Germany), counterstained with H33342 dye (H33342; Sigma-Aldrich, Merck KGaA, Darmstadt, Germany) and mounted with Vectashield (H-1000, Biozol Eching, Germany). Stained tissue sections were scanned with an AxioScan.Z1 digital slide scanner (Zeiss, Jena, Germany) equipped with a 20x magnification objective. The whole area of one section per liver was analysed. ICG-based signals were detected with a filter set FT 762 - BP785/38 and H33342 with a filter set FT 405 - BP 425/30. Images of the entire tissue sections were acquired and evaluated using the commercially available image analysis software Definiens Developer XD 2 (Definiens AG, Munich, Germany). The calculated parameter was the mean ICG fluorescence intensity of the whole liver tissue section.

## Statistics

Data were analysed using GraphPad Prism (v. 8.4.2; GraphPad Software, La Jolla, CA) except for receiver operating characteristics (ROC) analysis, which was performed using easyROC (v. 1.3.1; easyROC, www.biosoft.hacettepe.edu.tr/easyROC). All data presented as mean ± 95% CI unless elsewhere stated. Group size (n) is indicated for each experiment in figure legends. A Student's *t*-test was used to compare 2 independent groups. The Shapiro–Wilk test was used to assess the normal distribution of data. If the data did not pass the test, then the Mann–Whitney test was applied as an

> **The paper explained**
>
> **Problem**
> Obesity and non-alcoholic fatty liver disease (NAFLD) are worldwide health issues, and there is a pressing need for a sensitive, quantitative and non-invasive tool for assessing and monitoring the disease.
>
> **Results**
> We employed multispectral optoacoustic tomography (MSOT) to detect lipids in phantoms and in mouse tissues *in vivo* with high sensitivity. MSOT was used to differentiate grades of hepatic steatosis and quantitatively monitor lipid accumulation in mouse livers over time, without the use of contrast agents or labels. It was also used to track the real-time clearance kinetics in the liver of indocyanine green (ICG), which acts as a biomarker of liver function. Using MSOT, slower clearance of ICG was detected in a NAFLD mouse model, indicating a perturbed liver function.
>
> **Impact**
> The study establishes MSOT as an efficient imaging tool for the non-invasive, preclinical assessment of NAFLD, providing a foundation for longitudinal and therapeutic studies of the disease.

alternative to a *t*-test. The *z* test was used for comparisons of AUROC between 2 groups (DeLong *et al*, 1988). In ROC analyses, optimal cutoffs were determined using Youden's index. To express the monotonic relationship between measures, Spearman's correlation coefficient (*r*) with significance levels (*P*-values) was calculated. The correlation coefficient is defined as "strong" if *r* > 0.8 and as "excellent" if *r* > 0.9. A *P*-value of < 0.05 was considered as statistically significant. Significance was defined as \**P* < 0.05, \*\**P* < 0.01 and \*\*\**P* < 0.001.

# Data availability

The HE staining/anti-CD31 staining/ICG fluorescence images and part of the MSOT data from this publication have been deposited to the Zenodo database (https://zenodo.org). The assigned identifiers are as follows:

MSOT data: https://doi.org/10.5281/zenodo.4972949.
HE staining/anti-CD31 staining/ICG fluorescence images: https://doi.org/10.5281/zenodo.4975777.

The primary MSOT data for NAFLD mouse experiment, which is too large to be shared online (1.1 TB), are available from the authors upon request.

**Expanded View** for this article is available online.

# Acknowledgements

The authors would like to thank Sarah Glasl and Pia Anzenhofer for technical assistance, Doris Bengel for support in animal experiments and Robert J. Wilson for critical discussions on the manuscript. The research leading to these results has received funding from the Deutsche Forschungsgemeinschaft (DFG), Germany [Gottfried Wilhelm Leibniz Prize 2013; NT 3/10-1], from the European Research Council (ERC) under the European Union's Horizon 2020

research and innovation programme under Grant Agreement No. 694968 (PREMSOT) and from the Helmholtz-Gemeinschaft Deutscher Forschungszentren (HGF)/ExNet project "Innovative Intelligent Imaging" (i3-Helmholtz). Open Access funding enabled and organized by Projekt DEAL.

## Author contributions

SH conceived the study, designed and performed the experiments, analysed the data and wrote the manuscript. AB designed and performed the *ex vivo* experiments and analysed the data. AF designed and performed the *ex vivo* experiments and analysed the data. UK performed the *in vivo* experiments. RZT performed the *in vivo* experiments. SMH conceived and oversaw the animal study and provided advice for physiological data interpretation and manuscript composition. ACS conceived the study and wrote the manuscript. VN conceived the study and wrote the manuscript.

## Conflict of interest

V.N. is a shareholder of iThera Medical.

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
