## [Review Process File · EMBO Molecular Medicine]

Functional multispectral optoacoustic tomography imaging of hepatic steatosis development in mice

Shan Huang, Andreas Blutke, Annette Feuchtinger, Uwe Klemm, Robby Tom, Susanna Hofmann, Andre Stiel, and Vasilis Ntziachristos

DOI: [10.15252/emmm.202013490](https://doi.org/10.15252/emmm.202013490)

Corresponding author: Vasilis Ntziachristos (bioimaging.translatum@tum.de)

Review Timeline:

Submission Date:	22nd Sep 20
Editorial Decision:	17th Nov 20
Revision Received:	17th Jun 21
Editorial Decision:	1st Jul 21
Revision Received:	8th Jul 21
Accepted:	14th Jul 21

Editor: Lise Roth

Transaction Report:

17th Nov 2020

Dear Prof. Ntziachristos,

Thank you for the submission of your manuscript to EMBO Molecular Medicine, and please accept my apologies for the unusual delay in getting back to you. As mentioned in my previous communication, I initially only managed to secure two referees, and one of them never provided a report. Another reviewer then agreed to review the manuscript. However, given the contradictory reports that I got, and to reach a fair and balanced decision, I further sought advice from another expert in the field.

As you will see from the reports below, while referee #2 supports publication of the manuscript pending minor revisions, referee #4 raises a number of serious concerns (discussion, clarifications, lack of validation or quantification, overstatements), which should be convincingly addressed in a major revision of the present manuscript.

As mentioned above, I also sought advice from an independent expert, who stated: "The general hypothesis that MSOT can simply be used to detect lipids is highly questionable. The hypothesis should have been demonstrated/validated in simple objects first, such as tissue phantoms. This is an important step in studies like this because lipids (while abundant in the obese body) are very weak absorbers compared to all the other chromophores. Given that the linear spectral unmixing methods that underpin MSOT are severely limited in validity, it is doubtful that lipids can be quantified reliably in different animals. Interestingly, the authors themselves have acknowledged these limitations (Tzoumas et al) but seem to ignore these here. The authors simply assume that it will work in vivo, which is unlikely to be true."

Addressing the reviewers' concerns in full will be necessary for further considering the manuscript in our journal. As revising the manuscript according to the referees' recommendations appears to require a lot of additional work and experimentation, and given the potential interest of your findings, we are ready to extend the deadline to 6 months with the understanding that acceptance of the manuscript would entail a second round of review.

EMBO Molecular Medicine encourages a single round of revision only and therefore, acceptance or rejection of the manuscript will depend on the completeness of your responses included in the next, final version of the manuscript. For this reason, and to save you from any frustrations in the end, I would strongly advise against returning an incomplete revision. Should you find that the requested revisions are not feasible within the constraints outlined here and prefer, therefore, to submit your paper elsewhere, we would welcome a message to this effect.

When submitting your revised manuscript, please carefully review the instructions that follow below. Failure to include requested items will delay the evaluation of your revision:

1) A .docx formatted version of the manuscript text (including legends for main figures, EV figures

and tables). Please make sure that the changes are highlighted to be clearly visible.

2) Individual production quality figure files as .eps, .tif, .jpg (one file per figure).

3) A .docx formatted letter INCLUDING the reviewers' reports and your detailed point-by-point responses to their comments. As part of the EMBO Press transparent editorial process, the point-by-point response is part of the Review Process File (RPF), which will be published alongside your paper.

4) A complete author checklist, which you can download from our author guidelines (<https://www.embopress.org/page/journal/17574684/authorguide#submissionofrevisions>). Please insert information in the checklist that is also reflected in the manuscript. The completed author checklist will also be part of the RPF.

6) Before submitting your revision, primary datasets produced in this study need to be deposited in an appropriate public database (see <https://www.embopress.org/page/journal/17574684/authorguide#dataavailability>). Please remember to provide a reviewer password if the datasets are not yet public. The accession numbers and database should be listed in a formal "Data Availability " section (placed after Materials & Method). Please note that the Data Availability Section is restricted to new primary data that are part of this study.

7) We would also encourage you to include the source data for figure panels that show essential data. Numerical data should be provided as individual .xls or .csv files (including a tab describing the data). For blots or microscopy, uncropped images should be submitted (using a zip archive if multiple images need to be supplied for one panel). Additional information on source data and instruction on how to label the files are available at .

8) Our journal encourages inclusion of *data citations in the reference list* to directly cite datasets that were re-used and obtained from public databases. Data citations in the article text are distinct from normal bibliographical citations and should directly link to the database records from which the data can be accessed. In the main text, data citations are formatted as follows: "Data ref: Smith et al, 2001" or "Data ref: NCBI Sequence Read Archive PRJNA342805, 2017". In the Reference list, data citations must be labeled with "[DATASET]". A data reference must provide the database name, accession number/identifiers and a resolvable link to the landing page from which the data can be accessed at the end of the reference. Further instructions are available at .

9) We replaced Supplementary Information with Expanded View (EV) Figures and Tables that are collapsible/expandable online. A maximum of 5 EV Figures can be typeset. EV Figures should be cited as 'Figure EV1, Figure EV2" etc... in the text and their respective legends should be included in the main text after the legends of regular figures.

- For the figures that you do NOT wish to display as Expanded View figures, they should be bundled together with their legends in a single PDF file called *Appendix*, which should start with a

short Table of Content. Appendix figures should be referred to in the main text as: "Appendix Figure S1, Appendix Figure S2" etc.

- Additional Tables/Datasets should be labeled and referred to as Table EV1, Dataset EV1, etc. Legends have to be provided in a separate tab in case of .xls files. Alternatively, the legend can be supplied as a separate text file (README) and zipped together with the Table/Dataset file. See detailed instructions here:

10) The paper explained: EMBO Molecular Medicine articles are accompanied by a summary of the articles to emphasize the major findings in the paper and their medical implications for the non-specialist reader. Please provide a draft summary of your article highlighting

11) For more information: There is space at the end of each article to list relevant web links for further consultation by our readers. Could you identify some relevant ones and provide such information as well? Some examples are patient associations, relevant databases, OMIM/proteins/genes links, author's websites, etc...

12) Every published paper now includes a 'Synopsis' to further enhance discoverability. Synopses are displayed on the journal webpage and are freely accessible to all readers. They include a short stand first (maximum of 300 characters, including space) as well as 2-5 one-sentences bullet points that summarizes the paper. Please write the bullet points to summarize the key NEW findings. They should be designed to be complementary to the abstract - i.e. not repeat the same text. We encourage inclusion of key acronyms and quantitative information (maximum of 30 words / bullet point). Please use the passive voice. Please attach these in a separate file or send them by email, we will incorporate them accordingly.

Please also suggest a striking image or visual abstract to illustrate your article. If you do please provide a png file 550 px-wide x 400-px high.

13) As part of the EMBO Publications transparent editorial process initiative (see our Editorial at <http://embomolmed.embopress.org/content/2/9/329>), EMBO Molecular Medicine will publish online a Review Process File (RPF) to accompany accepted manuscripts.

In the event of acceptance, this file will be published in conjunction with your paper and will include the anonymous referee reports, your point-by-point response and all pertinent correspondence relating to the manuscript. Let us know whether you agree with the publication of the RPF and as here, if you want to remove or not any figures from it prior to publication.

EMBO Molecular Medicine has a "scooping protection" policy, whereby similar findings that are published by others during review or revision are not a criterion for rejection. Should you decide to

submit a revised version, I do ask that you get in touch after three months if you have not completed it, to update us on the status.

I look forward to receiving your revised manuscript.

Yours sincerely,

Lise Roth

Lise Roth, PhD
Editor
EMBO Molecular Medicine

To submit your manuscript, please follow this link:

Link Not Available

***** Reviewer's comments *****

Referee #2 (Remarks for Author):

The current manuscript describes non-invasive imaging and quantification of NAFLD development of HFD-fed mice by multispectral optoacoustic tomography (MSOT). The paper is easy to read, appears fairly novel, with or without labeling, and tackles with an important medical problem of

hepatic steatosis. After addressing the following rather minor points, I think the paper is suitable for publication in this journal.

1. There are a few studies investigating NAFLD models with photoacoustic-ultrasound hybrid imaging as cited in the paper (Ref #19,20). The authors should discuss their technological differentiation and advantages from the PA/US used in these papers.
2. The abbreviation "SV" is not described in the List of Abbreviations, and appears to be only accounted for in the figure legend.
3. Some part of References (i.e. Ref #24, 35, 38, 40) are in bold for the reason I do not know.
4. Fig 5A - I do not understand the labeling what #191 and #84 refer to.

Referee #4 (Remarks for Author):

The paper introduces an interesting new technology for evaluation of hepatic steatosis in mice. While the use of this approach should be of interest in the application, there are quite a substantial number of limitations to the existing study that diminish the potential impact of the work. The points below highlight both minor and major revisions that the reviewer believes could start to address these limitations, particularly with regard to data quantification and interpretation.

Introduction

P5 para 2: the statement that MSOT can directly visualise lipids based on characteristic absorption at 930 nm is misleading. Numerous biomolecules absorb light at the same wavelength, therefore the signal obtained with MSOT is not unique to lipids. The relative concentration of different biomolecules will also contribute to their extent of absorption. Thus the narrative in this paragraph should be reformulated to be scientifically precise.

Methods

P6 para "Animals"

- What are the overall number of animals used per group and how many yielded experimental data?
- How were animals caged?
- What gas was used to deliver anaesthesia for the experiments (e.g. O₂, medical air)?
- How long was the overall anaesthesia duration of the experiment?
- What was the maintained physiological status of the animals during experiment e.g. respiration / heart rates?
- ICG dose is quoted in microg/g, which is presumably /g animal body weight. As the obese mice are likely to have a very different body weight, could the authors please include here the average body weight in the two groups at the experimental time point?
- What was the study design in respect of the MSOT - histopathology analysis, are the same mice used for both studies?

P6 para "MSOT"

- Are the imaging wavelengths stated applicable to both the in vivo and ex vivo experiments? How many frame averages were used?

- Was any couplant used between the polyethylene membrane and the animal?
- What was the influence of using water in the water bath at the longer wavelengths of measurement? Describe the appearance of the background signal.

P7 para "Histopathology". No details are given of any MSOT - histopathology co-registration. How many sections are evaluated per mouse and how many fields per section? Are subsequent evaluations then made on a "per mouse" basis? Detail your biological and technical replicates in the methods section here as well as explicitly stating in the caption of each figure.

P7 Para "Oil-red-O staining". Same questions as above. Also, are livers from the same animals used for imaging divided into the FFPE and frozen parts for ex vivo analysis?

P8 para "Data analysis" Why was the recently reported eMSOT method of the same group not applied here? What is meant by "For detection of lipid, difference unmixing was also applied"? Oxygen saturation is not precisely sO₂ as measured directly in blood, suggest making that clear in the methods and results.

P8 para "Ex vivo quant.." Again, are these the same livers as the other parts but divided? Or separate animal cohorts? How many sections per liver were analysed.

Results

P9 para "MSOT imaging..."

It's rather unusual to undertake such quantitative analysis without first validating in phantoms the sensitivity of the imaging method to the respective analytes, especially given the complexity and heterogeneity seen in tissue and disease. The finding of no difference between ex vivo and in vivo tissues suggests challenges with the method of quantification, given that the change in oxygenation between in vivo and ex vivo environment is usually detectable by optical methods.

Fig. 1. Clarify that these images are taken from healthy mice in the figure caption. What are the min / max au values of the z color scale bars in (a)? Please label appropriately. Same applies to all subsequent images shown. No error bars are shown in (c), were these single experiments? Box and whisker plots in (d) and (e) are not very informative about the overall distribution and significance of the experiments, at least show the individual data points as has been done for the graphs in Fig. 2. The kidney and liver regions of interest cover only a small portion of the organ - this should be explained and justified.

Why normalize the tissue TBV and sO₂ to the aorta readouts? Were the lipid data normalized in any way? How do the AU readouts in each linear unmixing channel relate to actual concentrations in the tissue - linear / nonlinear etc?

Did you look at the portal vein itself compared to broader liver tissue (cf comment on venous blood in the organ)?

This reviewer is not convinced that the data presented are sufficient to support the concluding statement of this paragraph, which is rather over-reaching.

P10 para "MSOT imaging of steatosis..."

Again, stating that the ability of MSOT to differentiate key tissues was "confirmed" is not entirely accurate, as no validation was performed in independent animal cohorts to show that the measured spectra are truly characteristic of the specific organs shown.

No change in Hb or HbO₂ was observed in tissues where an increase in lipid deposit was shown. Is this not unexpected given the different vascularization in fatty vs non-fatty tissue?

Fig. 2. It would be helpful to label the organs visualised in (a). How were the regions of interest drawn for quantification in (C) and (D)? Also, there is quite a substantial heterogeneity seen in the MSOT signal from the different organs shown in (c) in both healthy and diseased, is this recapitulated in the underlying ex vivo IHC and other analyses? Is it technical or biological? The lack of reference data from the various ex vivo analyses presented in figures referred to makes it hard to rationalise this.

P10 para "Specific and sensitive ..."

The analysis conducted to verify the ability to detect lipids is only conducted in one mouse of each category and is severely limited in extent. Per-liver correlation between in vivo and ex vivo data would be necessary in this scenario to provide any confidence in the argument that is being made.

Does the difference quantification correlate with the linear unmixing result for lipid? The dynamic range of the difference data presented (Fig. 3E) is way smaller compared to the linear unmixing result so how is the use of the second metric justified in terms of analysing disease progression? Also appear to swap sensitivity for specificity in the ROC analysis, worth a comment.

The grading analysis is a good start, but again, should be reported as a correlation per animal rather than as a relative change between grades.

Was any analysis done on new and unseen animal cohorts generated at a different time point?

Fig. 3: Lacking any quantification of Oil-red-O across the cohorts. Please include and correlate to MSOT data.

Fig. 4: Please correlate the differences to MSOT data, the index evaluation from H&E is very different in its dynamic range compared to the MSOT output. Consider using a log scale in (f) to aid visualisation and include appropriate statistical analysis for longitudinal tracking of livers (and add longitudinal information from healthy mice as well).

P12 para "Functional OA imaging..."

Visualisation of the ICG inflow and clearance was surprisingly superficial in the supplied images and movie. Please explain. Assume quantification is made again in a small region of interest to extract the data shown?

Fig. 5: Can any quantification of the data in (e) be provided?

Discussion

Clinical translation is alluded to but potential limitations in this regard arising from the results shown in the paper or practical considerations are not addressed. Please provide a more balanced discussion of the findings of the paper, including its limitations. Also more broadly address the potential challenges to be encountered in translation beyond the relatively minor issues that are

already mentioned.

Point-by-Point Response to Reviewers' comments

Referee #2 (Remarks for Author):

The current manuscript describes non-invasive imaging and quantification of NAFLD development of HFD-fed mice by multispectral optoacoustic tomography (MSOT). The paper is easy to read, appears fairly novel, with or without labeling, and tackles with an important medical problem of hepatic steatosis. After addressing the following rather minor points, I think the paper is suitable for publication in this journal.

Point 1. There are a few studies investigating NAFLD models with photoacoustic-ultrasound hybrid imaging as cited in the paper (Ref #19,20). The authors should discuss their technological differentiation and advantages from the PA/US used in these papers.

We thank the reviewer for the comment. We have added a discussion on this point in the introduction section in the manuscript:

“Optoacoustic (OA) measurements (also known as photoacoustic) have been previously considered for investigating steatosis in vivo, but at the short wavelength infrared (SWIR) range, in particular at 1220 and 1370nm (1) (2). These studies only allowed the generation of bulk signals with a pixelated (speckle) appearance, that did not offer an anatomical optoacoustic reference, since the wavelengths employed do not reveal haemoglobin contrast. Moreover, generally, attenuation of light in the >1200nm region is higher to significantly higher than in the NIR. In contrast, our investigation herein focused on examining whether the 930nm range would be appropriate for imaging liver lipids and steatosis. Imaging in the NIR comes with several advantages as it can be seamlessly integrated with morphological and functional images obtained at the 700nm-900nm range afforded by the same tunable laser, and can yield higher sensitivity and specificity for the detection and quantification of lipids, by avoiding the strong water absorption in the >1200nm range..”

Point 2. The abbreviation "SV" is not described in the List of Abbreviations, and appears to be only accounted for in the figure legend.

We thank the reviewer for noticing. We have added SV to the list of Abbreviations in the manuscript.

Point 3. Some part of References (i.e. Ref #24, 35, 38, 40) are in bold for the reason I do not know.

We have fixed this issue.

Point 4. Fig 5A - I do not understand the labeling what #191 and #84 refer to.

We apologize for the confusion, and have changed the labeling on Fig 5A.

Referee #4 (Remarks for Author):

Point 1- The paper introduces an interesting new technology for evaluation of hepatic steatosis in mice. While the use of this approach should be of interest in the application, there are quite a substantial number of limitations to the existing study that diminish the potential impact of the work. The points below highlight both minor and major revisions that the reviewer believes could start to address these limitations, particularly with regard to data quantification and interpretation.

We thank the reviewer for acknowledging the novelty of our research and for the critical comments on the limitations of this work. To address these limitations, we have carried out a number of experiments and data analyses, and have adjusted the contents of the manuscript accordingly. These changes are described and discussed in the following point-to-point response.

Introduction

Point 2 - P5 para 2: the statement that MSOT can directly visualise lipids based on characteristic absorption at 930 nm is misleading. Numerous biomolecules absorb light at the same wavelength, therefore the signal obtained with MSOT is not unique to lipids. The relative concentration of different biomolecules will also contribute to their extent of absorption. Thus the narrative in this paragraph should be reformulated to be scientifically precise.

We thank the reviewer for this thoughtful comment. Indeed, there are various absorbers in the tissue that can absorb at this specific wavelength. What enables lipid detection by MSOT are the

absorption features of lipids in the near-infrared range; specifically, the characteristic peak at ~930nm (3). We have reformulated the paragraph to be more precise:

“Multi-spectral optoacoustic tomography (MSOT) separates lipids from other tissue components based on their absorption spectra in the near-infrared (NIR) range. Lipid spectra have a characteristic peak at a narrow spectral range around approximately 930 nm.”

Methods

P6 para "Animals"

Point 3- What are the overall number of animals used per group and how many yielded experimental data?

For the NAFLD development experiment (without ICG), 14 animals per sex were used in each group, for a total of 56 animals (28 males, 28 females). 48/56 animals yielded experimental data. For the ICG experiment, there were 6 animals in each group, in total 12 animals. 10/12 animals yielded experimental data. If health issues were noted during the experiment or discovered later by dissection, then these mice were excluded from the analysis.

We have now added the following to the Animals section of the Materials and Methods:

“14 mice per sex were used in each group (56 mice in total). 48 out of 56 animals yielded experimental data.”

“6 mice per group were used for the ICG study. 10 out of 12 animals yielded experimental data.”

Point 4- How were animals caged?

We have now added the following to the Animals section of the Materials and Methods:

“2-4 mice were kept in each cage.”

Point 5- What gas was used to deliver anaesthesia for the experiments (e.g. O₂, medical air)?

O₂ was used to deliver anesthesia for the experiment with a flow rate of 0.8L/min.

We have now adjusted the following sentence in the Multi-spectral Optoacoustic Tomography section of the Materials and Methods:

“For in vivo measurements, mice were anesthetized by continuous inhalation of 2% isoflurane (vaporized in 100% oxygen at 0.8 L/min) and subsequently placed within an animal holder in a supine position relative to the transducer array.”

Point 6- How long was the overall anaesthesia duration of the experiment?

We have now added the following to the Multi-spectral Optoacoustic Tomography section of the Materials and Methods:

“The total duration of anaesthesia was about 1 hour for the NAFLD experiment and about 2.5 hours for the ICG experiment.”

Point 7- What was the maintained physiological status of the animals during experiment e.g. respiration / heart rates?

The MSOT machine used in this study does not have an integrated respiration/heart rate monitoring system. However, the respiration rate was manually monitored by observing the live video. During the imaging, the respiration rate was kept in the range of 40-60 per minute.

We have now added the following to the Multi-spectral Optoacoustic Tomography section of the Materials and Methods:

“The respiration rate of the mouse was monitored during imaging and kept in the range of 40-60 per minute.”

Point 8- ICG dose is quoted in microg/g, which is presumably /g animal body weight. As the obese mice are likely to have a very different body weight, could the authors please include here the average body weight in the two groups at the experimental time point?

Indeed the two groups have different average body weights. The average body weight of the chow and HFD groups are 36.6g and 46.5g, respectively.

We have now added the following to the Animals section of the Materials and Methods:

“The average weights of the healthy and NAFLD mice were 36.6 g and 46.5 g, respectively.”

Point 9- What was the study design in respect of the MSOT - histopathology analysis, are the same mice used for both studies?

The same mice were used for both MSOT and histopathological analysis.

We have now added the following to the Animals section of the Materials and Methods:

“The same mice were used for imaging and histopathological analysis. After the final imaging time point, the imaged animal was sacrificed and the liver was isolated and divided into formalin-fixed, paraffin-embedded (FFPE) and frozen parts for ex vivo analysis.”

P6 para "MSOT"

Point 10- Are the imaging wavelengths stated applicable to both the in vivo and ex vivo experiments? How many frame averages were used?

Yes, the imaging wavelengths stated are applicable to both the *in vivo* and *ex vivo* experiments. 10 frame averages were used.

We have now added the following to the Multi-spectral Optoacoustic Tomography section of the Materials and Methods:

“For both in vivo and ex vivo measurements, imaging was performed at 27 wavelengths in the range of 700 nm- 960 nm in steps of 10 nm. At each wavelength, 10 frames were averaged per section during data acquisition.”

Point 11- Was any couplant used between the polyethylene membrane and the animal?

We have now added the following to the Multi-spectral Optoacoustic Tomography section of the Materials and Methods:

“Ultrasound gel (Ultrasound gel AQUASONIC clear, Parker Laboratories, Fairfield, USA) was used as a couplant between the polyethylene membrane and the animal.”

Point 12- What was the influence of using water in the water bath at the longer wavelengths of measurement? Describe the appearance of the background signal.

Water as a coupling medium absorbs minimal light at wavelengths shorter than 900 nm, but it does indeed absorb more significantly at longer wavelengths. Some studies have indicated that using water as coupling medium affects the sensitivity when detecting absorbers such as lipid, collagen, sugar and proteins, as the fluence of light at longer wavelengths is diminished by the water before reaching the sample (4). However, we deem the water absorption as not influential for lipid analysis, focusing on the lipid absorption peak around 930 nm.

To address this concern, we have now carried out two phantom experiments to verify the sensitivity of lipid detection, which are shown in Fig EV2,3. We observed in the imaging background flat, low signals from 700 nm to 930 nm and elevated signals from 930 nm to 960 nm (Appendix

Figure S3). The signal intensity of the background throughout the entire wavelength range remained relatively low compared to the imaging subjects.

Point 13

P7 para "Histopathology". No details are given of any MSOT - histopathology co-registration. How many sections are evaluated per mouse and how many fields per section? Are subsequent evaluations then made on a "per mouse" basis? Detail your biological and technical replicates in the methods section here as well as explicitly stating in the caption of each figure.

For MSOT analysis, the apical region of the right lateral lobe was always used as the ROI. Histopathology was carried out on random liver regions. Although the percentage of fat is not completely homogenous throughout all liver lobes, which was confirmed by histopathological quantification in our study, the steatosis grades remained the same (Appendix Figure S4). This heterogeneity was also reported in the literature (5). Since we employed semi-quantitative grading as the validation assay in our study design, we did not co-register MSOT with histopathology. For histopathology, the whole field from one liver section (average area: 35.31 mm²) is evaluated per mouse.

We have now added the following to the Multi-spectral Optoacoustic Tomography section of the Materials and Methods:

"Histopathology was carried out on random liver regions. One slice (average area: 35.31 mm²) per liver was analysed for each animal."

We have also added the following to the 'Ex vivo quantification of ICG tracer' section of the Materials and Methods:

"The whole area of one section per liver was analysed."

We have also detailed the replicates in the methods section and in the caption of each figure.

Point 14

P7 Para "Oil-red-O staining". Same questions as above. Also, are livers from the same animals used for imaging divided into the FFPE and frozen parts for ex vivo analysis?

Yes, the liver from the same mouse used for imaging was divided into FFPE and frozen parts for ex vivo analysis. This information has now been added into the Animal part in Materials and Method section:

“The same mice were used for imaging and histopathological analysis. After the final imaging time point, the animal was sacrificed and the liver was isolated and divided into FFPE and frozen parts for ex vivo analysis.”

Oil-red-O staining was only employed on one animal with severe steatosis for a sharp demonstration of excessive fat in liver, as it shows a bright pink color for lipid. However, there are technical challenges for quantifying lipid precisely with this type of staining, as lipid tends to loosen and wash away during the process, leading to imprecise quantification result due to the technical variance. Therefore for quantification of fat, we used an algorithm-based automated quantification on HE staining (6). A similar quantification method has been employed for the quantification of liver fat and correlation to MRI data (7). The description of the quantification method has now been added in to the Histopathology section in Materials and Methods:

“Additionally, the amount of lipid-vacuoles in the liver sections were morphometrically determined by automatic digital image analysis of scanned slides (26) (Axio Scan.Z1 scanner equipped with 20x objective, Zeiss, Germany), using the commercially available software Definiens Developer XD 2 (Definiens AG, Germany). Entire liver sections with an average area of 40 mm² were analysed. A specific rule set was defined in order to detect and quantify the lipid vacuoles of the hepatocytes based on morphology, size, pattern, shape, neighbourhood and special colour features. The calculated parameter was the percentage of surface areas considered as lipid vacuoles, divided by the total surface area of the whole analysed tissue section for each slide.”

P8 para "Data analysis"

Point 15 - Why was the recently reported eMSOT method of the same group not applied here?

eMSOT was designed to improve the accuracy of sO₂ measurements in deep tissue by addressing the spectral collapse problem. However, the algorithm of eMSOT only considers contributions from blood at wavelengths from 700 nm to 900 nm. It has only been tested in muscle and tumor – tissues that lack lipid. When applying eMSOT in a lipid-rich tissue, such as adipose tissue or steatotic liver, lipid will contribute to the signal from 860 nm to 900 nm, which will affect the readout of sO₂. Therefore, it is not suitable to employ eMSOT to analyze tissue with fat content.

Point 16 - What is meant by "For detection of lipid, difference unmixing was also applied"?

“Difference” is a data analysis method based on simple calculations of the differences between the readouts from two wavelengths. Although this method does not decompose the measured spectra into a collection of constituent spectra, it is still listed as an unmixing method in the software used for data analysis used in this study. We modified this method by further dividing the difference

value by the intensity at 800nm for normalization. We now refer to this method as “Difference analysis” to avoid confusion.

We have modified the content in the ‘Data analysis’ section of the Materials and Methods to clarify this point:

“For lipid detection, an additional method was also applied, termed ‘difference analysis’, which is based on a simple calculation of readout from 700, 800, and 930 nm (see more detail in Result section ‘Specific and sensitive detection and quantification of lipid in liver by MSOT’).”

We have further modified the content in the ‘Specific and sensitive detection and quantification of lipid in liver by MSOT’ section of the Results:

“In addition to the linear unmixing method, we introduce a simpler analysis method for label-free lipid detection, referred to herein as “difference analysis”. The readout from this method is calculated by subtracting the optoacoustic intensity at 930 nm from that at 700 nm and dividing that value by the intensity at 800 nm (difference analysis readout = $I_{700}-I_{930}/I_{800}$). Since the intensity at 930 nm varies with lipid content, while the value at 700 nm stays relatively constant because it is related to blood content, the difference value $I_{700}-I_{930}$ should be inversely proportional to lipid concentration. The intensity at 800 nm was used to normalize the difference value $I_{700}-I_{930}$ because oxygenation saturation does not affect the readout from this wavelength as it is the oxy-deoxy isobestic point. The main advantage of using such a simple readout for lipid content is that it allows a fast data acquisition and analysis as well as an economic light source with only three wavelengths.”

Point 17 - Oxygen saturation is not precisely sO₂ as measured directly in blood, suggest making that clear in the methods and results.

The term ‘oxygen saturation’ we used in this study could be more precisely referred to as blood oxygen saturation levels. It is measured and calculated from the blood in the analyzed tissue. It is different from the sO₂ widely assessed by the pulse oximeter, which only measures sO₂ from arterial blood.

We have modified the content in the Data Analysis section of the Materials and Methods to clarify this point:

“The data analysis additionally consisted of the calculation of the total blood volume (TBV = HbO₂ + Hb, displayed as arbitrary units) and the oxygen saturation of blood in the tissue (sO₂ = HbO₂ / TBV, displayed as percentage).”

Point 18 - P8 para "Ex vivo quant.." Again, are these the same livers as the other parts but divided? Or separate animal cohorts? How many sections per liver were analysed.

For the ICG experiment, we set up a cohort of 12 mice. This group was split in half such that 6 mice were fed either a chow or HFD diet for 6 months before the experiment. Only 5 of 6 mice per group yielded final data. After imaging, the mice were sacrificed and the liver was isolated and divided into FFPE and frozen parts for ex vivo analysis. The whole area of one section per liver was analyzed. The above information was added into the Animal section of the Materials and Methods.

“The mice used in the ICG experiment were injected with 2.5 µg/g body weight of ICG (ICG-Paulsion von Verdye 25 mg, Diagnostic Green, Germany and USA) in 100 µl saline intravenously through a catheter. 6 mice per group were used for the ICG study. 10 out of 12 animals yielded experimental data. The average weights of the healthy and NAFLD mice were 36.6 g and 46.5 g, respectively. The same mice were used for imaging and histopathological analysis. After the final imaging time point, the animal was sacrificed and the liver was isolated and divided into FFPE and frozen parts for ex vivo analysis.”

Results

P9 para "MSOT imaging..."

Point 19 - It's rather unusual to undertake such quantitative analysis without first validating in phantoms the sensitivity of the imaging method to the respective analytes, especially given the complexity and heterogeneity seen in tissue and disease.

We thank the reviewer for this comment. To address this concern, we have performed two phantom studies. First, we analyzed a lipid phantom with lipid fractions ranging from 0% to 100% (Fig EV2). MSOT lipid readouts showed excellent correlation with the lipid fractions (Spearman $r=1$, $P<0.0001$). Furthermore, we measured phantoms with homogenized liver and fat tissues mixed in different ratios. We tested liver tissue phantoms with lipid fractions up to 60%, as the lipid fraction in steatotic livers rarely exceeds 60% (5, 8) (Fig EV3). MSOT lipid unmixing readouts showed excellent correlation with the lipid fraction (Spearman $r=1$, $p<0.0001$).

However, we would like to point out that the phantom studies have limitations when correlating their results to the performance of MSOT for *in vivo* measurements. The development of hepatic steatosis, which is accompanied by obesity development, leads to heterogeneity in the overlying tissue of liver and a decrease in imaging depth, which cannot be mimicked by a simple phantom study. As light propagates from shallow to deep tissue, it is absorbed by the tissue's intrinsic chromophores. The light fluence distribution reaching a certain voxel is expected to change due to the wavelength-dependent light propagation through the overlying tissue, known as "spectral

coloring” (9). So far, no algorithm can perfectly solve this issue caused by tissue heterogeneity, which makes the technique’s sensitivity to lipid different from animal to animal, and even from ROI to ROI within the same animal. Considering these challenges, we do not propose that MSOT can provide the absolute concentration of lipid in tissue. However, MSOT still can detect the relative levels in lipid concentration in tissue and distinguish the different grades of steatosis, which is the gold standard in the clinic for diagnosis of hepatic steatosis.

We have added discussion of the new phantom experiments and figures to the Results:

“To test the feasibility of using MSOT to quantitatively isolate excessive lipid from liver tissue, we analyzed a phantom with lipid fraction ranges from 0% to 100% (Fig EV2). At 930 nm, the MSOT signal intensity increased with elevating lipid fraction in the phantom (Fig EV2A,B) (Spearman $r = 0.98$, $P < 0.0001$). Linear unmixing readouts of lipid showed excellent correlation with the lipid fraction (Fig EV2C) (Spearman $r = 1$, $P < 0.0001$). Furthermore, we analyzed phantoms with homogenized liver tissue and lipid mixed in different ratios. According to the histopathological quantification and MRI-PDFF data from other studies, lipid fractions in steatotic livers rarely exceed 60% (32, 33). We tested a series of liver tissue phantoms with lipid fractions from 0% to 60% (Fig EV3). The lipid signature absorption peak at 930 nm became more prominent with increasing lipid fraction in the phantom (Fig EV3A,B). The MSOT lipid readouts showed excellent correlation with the lipid fraction (Spearman $r = 1$, $p < 0.0001$; Fig EV3C).”

Point 20 - The finding of no difference between ex vivo and in vivo tissues suggests challenges with the method of quantification, given that the change in oxygenation between in vivo and ex vivo environment is usually detectable by optical methods.

We thank the reviewer for this comment, which encouraged us to discuss in more detail the difference in absorption spectra between ex vivo and in vivo tissues. In the wavelength range of 700 nm to 800 nm, mainly deoxygenated blood contributes to the absorption, while after the oxy-deoxy isobestic point (800 nm) the absorption from oxygenated blood becomes increasingly dominant. Therefore, the sO_2 from blood in the tissue affects the absorption spectra. As shown in Fig 1C, the ex vivo spectra of liver, iBAT and kidney have a sharper descending slope in general, which reflects the drop of sO_2 in these blood-rich tissues. rpWAT spectra remained largely the same, as rpWAT has little blood content compared to the other three tissues. In general, their spectra were not completely altered and the spectral signatures were preserved. For example, despite the difference in sO_2 , we can visualize the characteristic fat peak at 930 nm in iBAT both in vivo and ex vivo.

We have now added more explanation of the difference between the in vivo and ex vivo spectra in relation to sO_2 changes to the ‘MSOT imaging of liver, kidney, adipose tissues, and blood vessels in non-obese mice’ section of the Results:

“As shown in Fig 1C, the ex vivo spectra of liver, iBAT, and kidney have a sharper descending slopes in general, which reflects the drop of sO₂ in these blood-rich tissues. rpWAT spectra remained largely the same due to a low blood content compared to the other three tissues. In general, the spectral signatures of tissues, e.g. the characteristic lipid peak at 930 nm in iBAT and rpWAT, were mostly unaltered.”

Point 21- Fig. 1. Clarify that these images are taken from healthy mice in the figure caption. What are the min / max au values of the z color scale bars in (a)? Please label appropriately. Same applies to all subsequent images shown. No error bars are shown in (c), were these single experiments? Box and whisker plots in (d) and (e) are not very informative about the overall distribution and significance of the experiments, at least show the individual data points as has been done for the graphs in Fig. 2. The kidney and liver regions of interest cover only a small portion of the organ - this should be explained and justified.

We thank the reviewer for bringing our attention to these issues and helping us to improve our figure presentation. These images and data are indeed from healthy mice and we have now added this information to the figure caption.

The min / max au values of the z color scale bars are now provided in every figure caption.

There is no error bar in (c) because this panel's purpose is to show the clearest contrast between the in vivo and ex vivo spectra from the same animal. We have additional data from multiple healthy animals for liver and kidney, which has now been added to the Appendix, Figure S5. From this data, we could also visualize the difference between in vivo and ex vivo spectra of these tissues, which is due to the change in sO₂.

For panel (d) and (e) in Figure 1, we have now changed the plot into a dot plot. We also made this change for the graphs in Figure.2.

To select the ROI of tissues for quantitative analysis, we need to refer to spectral information in addition to anatomical information, as the spectral collapse in deep locations affects the unmixing results. The weakening of light in deep tissue can be seen clearly from the single wavelength image at 800 nm and the image with unmixing data for Hb. For ROI selection, we pre-select a rough ROI from the reconstructed image at 800 nm, which is the oxy-deoxy isobestic point. At this wavelength, blood is the dominant absorber and sO₂ does not affect the absorption. For a blood-rich tissue like the liver, the absorption at 800 nm should be relatively homogenous. When there is a dramatic drop of absorption within the pre-ROI, it indicates either a collapsed spectra or the true boundary of the tissue in that section. After selection of the pre-ROI, we adjust the ROI in the Hb unmixing image to ensure fidelity, based on the assumption that the unmixing readout from a precise ROI should reasonably reflect the expected absorber distribution. A blood rich tissue, such as the liver or kidney, should have a relative homogenous distribution of Hb. However, as we mentioned in the discussion section of the manuscript, the manual ROI selection method applied in

this study is not perfect because it is time-consuming and not completely objective, which gives rise to technical variances in data analysis. To eliminate the technical variance, each data point shown in this study was averaged from three ROIs per sample. As there is a need for a faster and more objective ROI selection method, in the future we would like to develop an AI-based data analysis approach, which contains an automated ROI selection for targeted tissues.

We have added some more explanation on ROI selection to the 'Data analysis' section in the Materials and Methods:

"For ROI selection, a preliminary ROI based on anatomical information was drawn on the image reconstructed from data at 800 nm (the oxy-deoxy isobestic point). The final ROI was fine tuned on the Hb unmixing image by evaluating whether the observed absorber distribution was reasonable based on prior knowledge of the target tissue; for example, a blood rich tissue such as liver or kidney should have a relatively homogenous distribution of Hb."

Point 22 - Why normalize the tissue TBV and sO₂ to the aorta readouts? Were the lipid data normalized in any way?

We thank the reviewer for raising these questions, which gave us the opportunity to elaborate in more detail on our methodology. As we mentioned in our response to the last question, light fluence reduces with tissue depth. Therefore, in the deep tissue, the spectrum is corrupted to the extent that it no longer gives an accurate unmixing readout. There is so far no algorithm that can tackle this issue perfectly, as the spectrum collapse cannot be corrected without taking account of the tissue heterogeneity, which is so complex that it cannot be predicted or simulated with an existing model. Therefore, for TBV and sO₂, we use an approach that was published by our group in Cell Metabolism in 2019 (10), in which the TBV and sO₂ readout from targeted tissues were normalized by the readout from an artery, whose content is pure blood, and thus the sO₂ should be 100% under normal conditions. The assumption is that the spectrum collapse should have the same effect on the artery and other tissues. However, in that study, the chosen artery was rather superficial, had a small cross-section, and was close to a zone (the upper 90 degree sector) with no transducer placed. For a more reasonable normalization, we chose the aorta, which sits at similar depths as our main targeted tissue (liver), enables a larger ROI selection, and resides in the lower middle zone of MSOT detection.

Although the spectral collapse should also affect the unmixing of lipid in deep tissue, we do not normalize lipid readout from the tissue of interest to any reference tissue because there is no tissue that has pure or constant detectable fat content in the animal model we used. We could estimate that the sensitivity of lipid detection in the liver in obese animals is lower than that in non-obese animals because of the expanded adipose tissue overlaying the liver and the increased imaging depth in obese animals. However, nearly all the lipid analyses that gave a positive readouts in the liver were from obese animals, since non-obese animals normally do not have hepatic steatosis and the lipid readout from healthy livers are mostly close to 0. The diminished sensitivity in obese

animals does not affect the monotonic relation between lipid fraction and MSOT lipid readout. As far as we know, there is no justified approach available to correct the lipid unmixing readout from tissue in vivo. In the future, an AI-based method might be able to tackle the spectral collapse in issue and further improve the precision of MSOT for lipid quantification. Using other coupling media, such as heavy water, can also eliminate the influence of water absorption and increase the sensitivity of lipid detection by MSOT.

Point 23 - How do the AU readouts in each linear unmixing channel relate to actual concentrations in the tissue - linear / nonlinear etc?

We thank the reviewer for this question. The AU readout of MSOT can be affected by imaging depth and tissue surroundings, which are different from subject to subject in vivo. Therefore, a linear correlation between AU readouts and the actual concentration in the tissue is not expected. What we observe in the newly performed phantom study is that fat concentration correlates with MSOT AU readout in a nonlinear fashion (Fig EV2, 3). However, there is an excellent monotonic relationship between these two measures.

Point 24 - Did you look at the portal vein itself compared to broader liver tissue (cf comment on venous blood in the organ)?

We have not analyzed the portal vein because it is too deep in the animal and, consequently, the light fluence is too low. Although we could sometimes resolve the anatomy of the portal veins, their spectra were collapsed and the signal intensities low, which led to a misleading unmixing results.

Point 25 - This reviewer is not convinced that the data presented are sufficient to support the concluding statement of this paragraph, which is rather over-reaching.

We thank the reviewer for this important comment, which allowed us to expand the discussion on how MSOT differentiates different tissues/organs and its advantages over other imaging modalities. As shown in our manuscript, we examined the characteristics of key tissues, including the liver, kidney, brown adipose tissue (BAT), white adipose tissue (WAT), vein, and artery from three aspects: anatomy, absorption spectra, and tissue contents, as analyzed by MSOT unmixing. Similar to conventional imaging modalities such as MRI, CT, or ultrasound, MSOT imaging allows visualization of the anatomy of the organ for tissue segmentation and differentiation. What sets MSOT apart is that it can also analyze the molecular contents of the tissue for characterization and differentiation. We have demonstrated that the molecular readouts from the key tissues in vivo reflect their biological properties: 1) Liver, kidney, BAT are blood-rich tissues while WAT is not. 2)

Adipose tissues (BAT and WAT) contain lipids. WAT is dominated by lipid while BAT is not. 3) Liver has lower sO₂ than kidney, as it has more venous blood than kidney. 4) sO₂ is lower in veins than in arteries. We also compare the in vivo and ex vivo spectra of the tissues, which are different due to the difference in tissue sO₂. Despite this difference, we could still see the signatures that mark the biological properties of the tissues, such as the lipid peak at 930 nm in BAT and WAT. Therefore, we conclude that MSOT can differentiate the key tissues both anatomically and on a molecular level. We only performed the validation assay for lipid in liver, as the focus of this study is hepatic steatosis. Moreover, it is impossible to preserve the in vivo oxygenation status of blood after sacrificing the animal for ex vivo validation. Therefore, it is difficult to validate the Hb and HbO₂ readout using ex vivo data. To be more scientifically rigorous, we adjusted the phrasing in the manuscript.

“These results indicated that MSOT can analyse the blood and fat contents of tissues such as liver to reveal their biological properties.”

P10 para "MSOT imaging of steatosis..."

Point 26 - Again, stating that the ability of MSOT to differentiate key tissues was "confirmed" is not entirely accurate, as no validation was performed in independent animal cohorts to show that the measured spectra are truly characteristic of the specific organs shown. No change in Hb or HbO₂ was observed in tissues where an increase in lipid deposit was shown. Is this not unexpected given the different vascularization in fatty vs non-fatty tissue?

We thank the reviewer for bringing up this point. To address this concern, we performed a anti-CD31 antibody staining on the same livers (Appendix Figure S6). The results show no significant difference in CD31+ area between ND and HFD group (P = 0.1541), indicating no changes in vessel density. In the literature, there are some findings about perturbation of liver perfusion correlated to hepatic steatosis in humans (11) (12) and rats (13) (14). However, we have not observed this phenomenon in the animal models used in our study. From our data, we have not observed a statistically significant change in blood content or vessel density in steatotic livers compared to normal livers. One possible reason is that in our study, the 3 month HFD-fed mice only developed NAFLD that is not associated with hepatic microvascular dysfunction. As in humans, vascular changes occur more in severe steatosis (15).

Point 27 - Fig. 2. It would be helpful to label the organs visualised in (a). How were the regions of interest drawn for quantification in (C) and (D)

We added labels for liver and kidney ROIs (A). The ROI selection method is as described in our response to point 21.

Point 28 - Also, there is quite a substantial heterogeneity seen in the MSOT signal from the different organs shown in (c) in both healthy and diseased, is this recapitulated in the underlying ex vivo IHC and other analyses? Is it technical or biological? The lack of reference data from the various ex vivo analyses presented in figures referred to makes it hard to rationalise this.

We thank the reviewer for raising these questions. The data presented in this figure are from the cohorts that received 3 months of either a normal or a high fat diet. In the high fat diet group, the degree of steatosis developed were not identical. This biological variance can also be seen in the body weight data (Fig 2B). To better address the reviewer's questions, we now show an ex vivo histology quantification of the area of fat in liver histology sections from the same cohort (Appendix Figure S7). The ex vivo quantifications of liver fat are heterogeneous, similar to the MSOT results. Moreover, even in different regions of the same liver, the liver fat readouts vary (Appendix Figure S4). Thus, this heterogeneity is, at least partially, due to biological variance. It depends on the pathological status of the tissue and the region analyzed. Apart from the biological variance, the technical variance can also not be excluded. To minimize the technical variance, we analyzed 3 sections per animal and the MSOT data presented in the paper are always the average of 3 technical replicates. As discussed in the answer to point 22, we normalized the TBV and sO₂ to further minimize the technical variance for these two readouts, and this leads to a reduced data heterogeneity in TBV and sO₂ data compared to the original Hb and HbO₂ data. Unfortunately, we cannot use the same strategy for lipid analysis due to the lack of suitable internal references. However, despite the variance, the lipid data still achieves statistical significance thanks to the dramatic change in liver lipid content, sufficient sample size, and the sensitivity of the method, which will be discussed more in the following answers.

Point 29 - P10 para "Specific and sensitive ..."

The analysis conducted to verify the ability to detect lipids is only conducted in one mouse of each category and is severely limited in extent. Per-liver correlation between in vivo and ex vivo data would be necessary in this scenario to provide any confidence in the argument that is being made.

We thank the reviewer for these comments. However, we have verified the ability of MSOT to detect lipids in multiple mice of each category. The sample size was 6 and 9 for control and steatotic groups, respectively. We first showed the MSOT image from only one pair of mice and their corresponding spectra to make the difference in the spectra more palpable to the reader (Fig. 3A-D). Afterwards we showed the unmixing data from the whole cohort. We have now additionally

correlated the ex vivo quantification of lipid from histology with MSOT lipid readout (Fig EV5A). The data shows strong correlation (Spearman $r = 0.82$, $P < 0.0001$). Please see the answers to point 30 for more detail and discussion.

Point 30 - Does the difference quantification correlate with the linear unmixing result for lipid? The dynamic range of the difference data presented (Fig. 3E) is way smaller compared to the linear unmixing result so how is the use of the second metric justified in terms of analysing disease progression? Also appear to swap sensitivity for specificity in the ROC analysis, worth a comment.

We thank the reviewer for these comments, which allowed us to further discuss the difference between the two methods. Linear unmixing is used in many studies and this function is integrated into the analysis software (ViewMSOT) that we use. It is a standard way to analyze several absorbers simultaneously. This method takes into account the whole spectra and the algorithm calculates the contribution from each absorber. Using a wide range of wavelengths, as we did in our study, enhances the accuracy of the data output, which correlates with the concentration/density of the absorbers. Unlike linear unmixing, the difference analysis used in our study is a customized and simplified method for lipid detection, based on the characteristic peak of lipid at 930 nm. It calculates an index from the absorption intensity at three wavelengths, 700nm, 800nm, and 930nm, with a much smaller dynamic range than linear unmixing. This method can detect lipid in mild to severe steatotic livers, but the readout does not correlate with the severity of NAFLD. To better address this question, we performed a correlation analysis for the linear unmixing readout and the difference analysis readout (Fig EV5B). Given the fundamental difference in the algorithm between these two unmixing methods, their readouts do not show strong correlation (Spearman $r = 0.55$, $P = 0.0001$). Therefore, we stated in the manuscript that the difference analysis method can be used for detection of steatosis but it is not suitable for quantitative analysis required for disease progression monitoring. However, with less demand in terms of light source and faster data acquisition and analysis compared to linear unmixing, it could still be potentially translated to a screening method for early detection of steatosis.

Point 31 - The grading analysis is a good start, but again, should be reported as a correlation per animal rather than as a relative change between grades.

We appreciate this comment from the reviewer. However, we have to politely disagree with the opinion that a correlation per animal will help validate the performance of the imaging method, for two main reasons:

1) Despite being semi-quantitative/categorical, the histology based grading system is still considered the gold standard due to its independence from any in vivo imaging constraints (16) (17). Most therapeutic studies currently use grade change as a reference for non-invasive

monitoring of steatosis due to its robustness. In fact, several recent MRI studies published in top journals in the hepatology field also correlated imaging results to histology/3-stage scale, despite the possibility to perform a per subject correlation of quantitative data (e.g. Middleton et al., Gastroenterology, 2017; Jayakumar et al., Journal of Hepatology, 2019 and Loomba et al., Hepatology, 2020) (18) (19) (20). MRI is valued as a non-invasive imaging method for steatosis quantification, and according to the EASL–EASD–EASO Clinical Practice Guidelines for the management of NAFLD (21) can be termed a “gold standard” for quantitative assessment of steatosis. Similar to MRI, our MSOT based method could deliver continuous numerical data on steatosis, but we argue that a continuous scale does not add true value to the staging of the disease if the pathophysiology and phenotype is reflected in the established grades.

2) For a valid correlation analysis, it is necessary to perform a strict co-registration of the imaging and a second quantitative assay, which is not feasible in our study. As we have discussed previously, the lipid density varied in different regions of the same liver, according to the results from quantitative analysis of histopathology. As shown in Appendix figure S4, the fat coverage can range from 10% to 30% in the same liver with identical steatosis grade, as the grade merely reflects the percentage of hepatocytes affected by lipid droplets without considering the size of the droplets. To better address this point, we did a quantitative analysis from histopathology and correlated the data to MSOT data (Fig EV5A). Although the data shows strong correlation (Spearman $r = 0.82$, $P < 0.0001$), we would not draw conclusions from this result because the scattering of the data points reflects the inconsistencies caused by lack of co-registration of ROIs. *Therefore, we rather propose lipid in liver measured by MSOT as a categorical biomarker to define the severity grade of steatosis with the numerical readout.* We have now added this discussion and the correlation data to the manuscript.

Point 32 - Was any analysis done on new and unseen animal cohorts generated at a different time point?

In this section, only the data in Figure 4E-F are from the same cohort of mice at different time points. All other data are grouped by the severity of steatosis according to histopathology grading rather than treatment time point. They are from mice at time points ranging from 1 to 6 month on a specific diet.

Point 33 - Fig. 3: Lacking any quantification of Oil-red-O across the cohorts. Please include and correlate to MSOT data.

We thank the reviewer for this comment. The purpose of showing Oil-red-O staining is to allow a straight-forward visualization of the lipid in the liver, which stains in red. However, according to our pathologist, Oil-red-O staining is not suitable for quantification of lipid because, although the lipid

can be stained efficiently, the fat stain tends to fall off during the staining process, which will affect the accuracy of the quantification. Therefore, we performed automated quantification of fat in liver on HE stainings, which is more accurate than quantification on Oil-red-O staining. The data is shown in Appendix figure S7.

Point 34 - Fig. 4: Please correlate the differences to MSOT data, the index evaluation from H&E is very different in its dynamic range compared to the MSOT output.

Please see our response to point 30.

Point 35 - Consider using a log scale in (f) to aid visualization and include appropriate statistical analysis for longitudinal tracking of livers (and add longitudinal information from healthy mice as well).

We thank the reviewer for the suggestion of using a log scale in (f). However, using a log scale will cause the loss of data points valued 0, which occurs often at the 1 month time point. Therefore, we kept the linear scale. We have now added longitudinal information from healthy mice to Fig.4.

Point 36 - P12 para "Functional OA imaging..."

Visualisation of the ICG inflow and clearance was surprisingly superficial in the supplied images and movie. Please explain. Assume quantification is made again in a small region of interest to extract the data shown?

We thank the reviewer for this comment. The ROIs were superficial because: 1) the sections we were imaging contain the apical region of the right lobe of the liver, which is superficial and thin; 2) although anatomical features of the liver might still be observable, deeper regions are not suitable for analysis, possibly due to the spectrum collapse effect that hinders unmixing. Therefore, to ensure the fidelity of the unmixing data, the final ROIs are superficial.

Point 37 - Fig. 5: Can any quantification of the data in (e) be provided?

The data is now provided in Appendix Figure S2.

Discussion

Point 38 - Clinical translation is alluded to but potential limitations in this regard arising from the results shown in the paper or practical considerations are not addressed. Please provide a more balanced discussion of the findings of the paper, including its limitations. Also more broadly address the potential challenges to be encountered in translation beyond the relatively minor issues that are already mentioned.

We thank the reviewer for this comment. We have now added a paragraph to the discussion section that more broadly addresses the limitations revealed in our study.

“Our study reveals some limitations of MSOT-based steatosis assessment. First, although the numeric readout of MSOT correlates to different grades of steatosis with proper thresholds, it cannot be converted into an absolute lipid fraction, despite its continuity. The main reasons for this are: 1) Heterogeneity of tissues and body composition in vivo. The unmixing readout is calculated from the absorption spectrum, which is greatly affected by the tissue that surrounds the ROI. Tissue contents and body composition vary from subject to subject, especially in the context of metabolic disease. This leads to an individualized problem that no unmixing algorithm can yet solve. In the future, developing an AI-based method could address this problem. 2) Lack of co-registration of MSOT ROI and sample ROI for the validation assay. Since MSOT can only analyse a limited volume in the liver due to the depth, a co-registration of MSOT and the validation assay that provides the ground truth lipid fraction readout is indispensable for the conversion of MSOT a.u. into absolute lipid fraction. It is challenging if the validation assay is an ex vivo method. One possibility is to co-register MSOT and MRI-PDF images. However, an advanced algorithm that solves the tissue heterogeneity problem discussed in 1) is still a prerequisite for this conversion. A second major limitation revealed in our study is the imaging depth, which is not sufficient for analysing a whole liver section. This is mainly due to the spectrum corruption in deep tissue. Since liver is rich in blood, a strong absorber for the wavelengths we used, the excitation light is quickly exhausted in liver tissue. Therefore we need to avoid selecting deep voxels with corrupted spectra by using a two-step method for ROI selection described in the Materials and Methods section. However this method is time-consuming and not completely objective. To improve the ROI selection for future studies, it would be useful to develop an automated method using deep learning, which could cluster spectra of specific tissues and correct the corrupted spectra until the signals drop beyond an auto-determined threshold. Another technical issue is water as coupling media could diminish the sensitivity of lipid detection by MSOT, as water absorbs at wavelengths longer than 900 nm. Using other coupling media, such as heavy water, can eliminate the influence of water absorption and increase the sensitivity of lipid detection by MSOT.”

However, we still believe that the major challenges to be encountered in clinical translation are those mentioned in the original manuscript, namely the difference in modality performance and the imaging depth. We have now added some contents to support these points. The limitations we discussed in the newly added paragraph will also affect the translation of our method. However

since they are general issues that already affect the performance of our preclinical method, we do not consider them as specific challenges for clinical translation.

1. Rom O, Xu G, Guo Y, Zhu Y, Wang H, Zhang J, Fan Y, et al. Nitro-fatty acids protect against steatosis and fibrosis during development of nonalcoholic fatty liver disease in mice. *EBioMedicine* 2019;41:62-72.
2. Xu G, Meng ZX, Lin JD, Deng CX, Carson PL, Fowlkes JB, Tao C, et al. High resolution Physio-chemical Tissue Analysis: Towards Non-invasive In Vivo Biopsy. *Sci Rep* 2016;6:16937.
3. Jacques SL. Optical properties of biological tissues: a review. *Phys Med Biol* 2013;58:R37-61.
4. Prakash J, Seyedebrahimi MM, Ghazaryan A, Malekzadeh-Najafabadi J, Gujrati V, Ntziachristos V. Short-wavelength optoacoustic spectroscopy based on water muting. *Proc Natl Acad Sci U S A* 2020;117:4007-4014.
5. Bannas P, Kramer H, Hernando D, Agni R, Cunningham AM, Mandal R, Motosugi U, et al. Quantitative magnetic resonance imaging of hepatic steatosis: Validation in ex vivo human livers. *Hepatology* 2015;62:1444-1455.
6. Sachs S, Niu L, Geyer P, Jall S, Kleinert M, Feuchtinger A, Stemmer K, et al. Plasma proteome profiles treatment efficacy of incretin dual agonism in diet-induced obese female and male mice. *Diabetes Obes Metab* 2020.
7. Raptis DA, Fischer MA, Graf R, Nanz D, Weber A, Moritz W, Tian Y, et al. MRI: the new reference standard in quantifying hepatic steatosis? *Gut* 2012;61:117-127.
8. Tang A, Tan J, Sun M, Hamilton G, Bydder M, Wolfson T, Gamst AC, et al. Nonalcoholic fatty liver disease: MR imaging of liver proton density fat fraction to assess hepatic steatosis. *Radiology* 2013;267:422-431.
9. Tzoumas S, Deliolanis N, Morscher S, Ntziachristos V. Unmixing Molecular Agents From Absorbing Tissue in Multispectral Optoacoustic Tomography. *IEEE Trans Med Imaging* 2014;33:48-60.
10. Reber J, Willershauser M, Karlas A, Paul-Yuan K, Diot G, Franz D, Fromme T, et al. Non-invasive Measurement of Brown Fat Metabolism Based on Optoacoustic Imaging of Hemoglobin Gradients. *Cell Metab* 2018;27:689-701 e684.
11. Rijzewijk LJ, van der Meer RW, Lubberink M, Lamb HJ, Romijn JA, de Roos A, Twisk JW, et al. Liver fat content in type 2 diabetes: relationship with hepatic perfusion and substrate metabolism. *Diabetes* 2010;59:2747-2754.
12. Ijaz S, Yang W, Winslet MC, Seifalian AM. Impairment of hepatic microcirculation in fatty liver. *Microcirculation* 2003;10:447-456.
13. Pereira E, Silveiras RR, Flores EEI, Rodrigues KL, Ramos IP, da Silva IJ, Machado MP, et al. Hepatic microvascular dysfunction and increased advanced glycation end products are components of non-alcoholic fatty liver disease. *PLoS One* 2017;12:e0179654.
14. Pasarin M, Abraldes JG, Rodriguez-Vilarrupla A, La Mura V, Garcia-Pagan JC, Bosch J. Insulin resistance and liver microcirculation in a rat model of early NAFLD. *J Hepatol* 2011;55:1095-1102.

15. Wanless IR, Shiota K. The pathogenesis of nonalcoholic steatohepatitis and other fatty liver diseases: a four-step model including the role of lipid release and hepatic venular obstruction in the progression to cirrhosis. *Semin Liver Dis* 2004;24:99-106.
16. Machado MV, Cortez-Pinto H. Non-invasive diagnosis of non-alcoholic fatty liver disease. A critical appraisal. *J Hepatol* 2013;58:1007-1019.
17. Fedchuk L, Nascimbeni F, Pais R, Charlotte F, Housset C, Ratzu V, Group LS. Performance and limitations of steatosis biomarkers in patients with nonalcoholic fatty liver disease. *Aliment Pharmacol Ther* 2014;40:1209-1222.
18. Jayakumar S, Middleton MS, Lawitz EJ, Mantry PS, Caldwell SH, Arnold H, Mae Diehl A, et al. Longitudinal correlations between MRE, MRI-PDFF, and liver histology in patients with non-alcoholic steatohepatitis: Analysis of data from a phase II trial of selonsertib. *J Hepatol* 2019;70:133-141.
19. Loomba R, Neuschwander-Tetri BA, Sanyal A, Chalasani N, Diehl AM, Terrault N, Kowdley K, et al. Multicenter Validation of Association Between Decline in MRI-PDFF and Histologic Response in NASH. *Hepatology* 2020;72:1219-1229.
20. Middleton MS, Heba ER, Hooker CA, Bashir MR, Fowler KJ, Sandrasegaran K, Brunt EM, et al. Agreement Between Magnetic Resonance Imaging Proton Density Fat Fraction Measurements and Pathologist-Assigned Steatosis Grades of Liver Biopsies From Adults With Nonalcoholic Steatohepatitis. *Gastroenterology* 2017;153:753-761.
21. European Association for the Study of the L, European Association for the Study of D, European Association for the Study of O. EASL-EASD-EASO Clinical Practice Guidelines for the management of non-alcoholic fatty liver disease. *J Hepatol* 2016;64:1388-1402.

1st Jul 2021

Dear Prof. Ntziachristos,

Thank you for the submission of your revised manuscript to EMBO Molecular Medicine. We have now received the report from the expert who reviewed your revised manuscript and your responses to the referees' comments. This referee had already seen and provided advice on the initial version of your manuscript. As you will see, this referee is now supportive of publication, and I am therefore pleased to inform you that we will be able to accept your manuscript once the following editorial points will be addressed:

1/ Main manuscript text:

- Please address the minor edit in track changes mode suggested by our data editors in the main manuscript file labelled 'Related manuscript file'. Please use this file for any further modification.
- Please remove the highlighted text, and only keep in track changes the new modifications.
- Please reorder the different sections so as to have Introduction, Results, Discussion, Material and Methods, Data availability, Acknowledgements, Author Contributions and Competing Interests Statement (which should be renamed Conflict of Interest).
- Please remove the abbreviation list, and instead incorporate the abbreviations in the manuscript text.
- Thank you for providing a Data Availability section. For imaging data, we recommend Image Data Resource (IDR) but only certain types of datasets are accepted. If IDR is not an option, we recommend Biostudies, but zenodo is fine too. However, you could put your 1.1TB data on Biostudies if you wish to.
- Please merge the funding information with the Acknowledgements (after the Data Availability section).
- Please update the reference format so as to have them in alphabetical order, and with 10 authors listed before et al.
- Please remove the Table 1 legend and add it to the table.
- Please add a heading "Expanded View Figure Legends" after the main figure legends.
- Please remove the Appendix and movie legends from the main manuscript file.

2/ Figures and Appendix:

- Please indicate in the main and appendix figures or in their legends the exact p= values, not a range, along with the statistical test used.
- Appendix: please correct the nomenclature to "Appendix Figure S1" etc.
- Movie: please correct the nomenclature to "Movie EV1". The legend should be zipped with the movie file.
- Please make sure that all figures are referenced to in the manuscript text, and in the chronological order (currently Fig. EV1 is called out after Fig. EV3 and the Appendix figures S3-7 are not called out).

3/ Checklist:

Please fill out the author information (top left corner), and provide information in Section 1a, 6 (references of antibodies) and 18.

4/ Thank you for providing The paper explained. I slightly modified the text, please let me know if

you agree with the following and include it in the main manuscript file:

Problem

Obesity and non-alcoholic fatty liver disease (NAFLD) are worldwide health issues and there is a pressing need for a sensitive, quantitative and non-invasive tool for assessing and monitoring the disease.

Results

We employed multispectral optoacoustic tomography (MSOT) to detect lipids in phantoms and in mouse tissues in vivo with high sensitivity. MSOT was used to differentiate grades of hepatic steatosis and quantitatively monitor lipid accumulation in mouse liver over time, without the use of contrast agents or labels. It was also used to track the real-time clearance kinetics in the liver of indocyanine green (ICG), which acts as a biomarker of liver function. Using MSOT, slower clearance of ICG was detected in a NAFLD mouse model, indicating a perturbed liver function.

Impact

The study establishes MSOT as an efficient imaging tool for the non-invasive, preclinical assessment of NAFLD, providing a foundation for longitudinal and therapeutic studies of the disease.

5/ Thank you for providing a synopsis text. I included minor modifications, please let me know if you agree with the following:

Multispectral optoacoustic tomography (MSOT) is demonstrated for the first time as an efficient imaging tool for the non-invasive assessment of hepatic steatosis in a non-alcoholic fatty liver disease (NAFLD) mouse model.

- MSOT allows visualization and quantification of lipids without labels or contrast agents.
- MSOT allows non-invasive detection and distinction between different grades of hepatic steatosis.
- MSOT allows quantitative monitoring of lipid accumulation in the liver over time.
- MSOT allows real-time tracking of clearance kinetics of indocyanine green in the liver to evaluate liver function and assess NAFLD severity.

6/ As part of the EMBO Publications transparent editorial process initiative (see our Editorial at <http://embomolmed.embopress.org/content/2/9/329>), EMBO Molecular Medicine will publish online a Review Process File (RPF) to accompany accepted manuscripts.

This file will be published in conjunction with your paper and will include the anonymous referee reports, your point-by-point response and all pertinent correspondence relating to the manuscript. Let us know whether you agree with the publication of the RPF.

I look forward to receiving your revised manuscript.

Yours sincerely,

Lise Roth

Lise Roth, PhD
Editor
EMBO Molecular Medicine

To submit your manuscript , please follow this link:

Link Not Available

***** Reviewer's comments *****

Referee #3 (Remarks for Author):

The authors addressed critical questions that arose in the last review by providing more detailed information and additional measurements. Important questions regarding the capability of the method to quantify the lipid content in vivo in small animals were answered and, importantly, the limitations the method were discussed. The publication of the manuscript is recommended.

The authors performed the requested editorial changes.

14th Jul 2021

Dear Prof. Ntziachristos,

Thank you for sending the synopsis picture. We are pleased to inform you that your manuscript is now accepted for publication and will be sent to our publisher to be included in the next available issue of EMBO Molecular Medicine.

Congratulations on your interesting work!

With kind regards,

Lise Roth

Lise Roth, Ph.D
Editor
EMBO Molecular Medicine

Follow us on Twitter @EmboMolMed
Sign up for eTOCs at embopress.org/alertsfeeds

Corresponding Author Name: Vasilis Ntziachristos

Manuscript Number: EMM-2020-13490